# KA2L: A Knowledge-Aware Active Learning Framework for LLMs

## Abstract

Fine-tuning large language models (LLMs) with high-quality knowledge has been shown to enhance their performance effectively. However, there is a paucity of research on the depth of domain-specific knowledge comprehension by LLMs and the application of targeted active learning to improve their expertise. To address this gap, we introduce the **Knowledge-Aware Active Learning (KA2L)** framework. This framework assesses LLMs' mastery of specific knowledge points to aid in constructing unanswerable or unknowable questions through latent space analysis. This active learning strategy enhances training efficiency by focusing on knowledge the model has yet to master, thereby minimizing redundancy in learning already acquired information. This study innovatively employs a knowledge distribution probing technique to examine the hidden states of specific Transformer layers and identify the distribution of known and unknown knowledge within the LLM. Additionally, a hidden-state decoding method is proposed to generate numerous unknown questions in natural language from the latent knowledge space. In our experiments, we selected nine open-source LLMs to validate the effectiveness of the proposed framework. Results indicate that KA2L not only significantly reduces 50% annotation and computation costs across two open-domain and one vertical-domain dataset but also achieves better performance, offering valuable insights into active learning strategies for LLMs. The code is available at https://anonymous.4open.science/r/KA2L-F15C.

## 1 Introduction

Large Language Models (LLMs) such as GPT-4 and Llama-3 have demonstrated remarkable capabilities across a wide range of NLP tasks (Zhao et al., 2025), and there is a growing demand for applying them to specific domains. Enhancing the domain-specific knowledge of LLMs primarily relies on techniques such as Supervised Fine-Tuning (SFT) (Zhao et al., 2025) and Retrieval Augmented Generation (RAG) (Gao et al., 2024). These methods typically require substantial amounts of high-quality annotated data or external knowledge bases. However, in practical applications, two significant challenges arise: (1) The knowledge mastered by LLMs is often invisible, making it necessary to train on the entire domain knowledge during each SFT process, leading to significant resource waste. (2) Without visibility into the model's knowledge, it is difficult to explicitly ascertain the new knowledge required by the LLM. This results in a "black-box" learning process, where incremental learning of new knowledge is prone to noise due to the typically low proportion of new knowledge in the overall training set, ultimately affecting the model's learning efficiency. Therefore, this paper proposes a novel active learning framework that focuses on detecting the distribution of known and unknown knowledge within LLMs. By directing training toward under-learned or unknown knowledge, the framework avoids redundant annotation and repetitive learning on already-acquired concepts, enabling more efficient and targeted knowledge acquisition.

Traditional active learning (AL) aims to identify a small subset of high-value samples from a large data pool, allowing a model to approximate the performance attainable with full-dataset training. Prominent strategies include uncertainty-based, diversity-based, and gradient-based approaches. For instance, diversity-based methods like Coreset (Sener & Savarese, 2018) select samples that are maximally different from one another to enhance model generalization. Hybrid approaches such as BADGE (Ash et al., 2020) leverage gradients on models like ResNet (He et al., 2016), embodying the core principle of selecting samples based on a combination of uncertainty and diversity. How-

ever, directly applying these methods to modern LLMs is challenging due to prohibitive computational costs and a paradigm mismatch, as they were primarily designed for classification rather than generative tasks. Consequently, current AL research for LLM has largely shifted toward distillation-based (e.g. FreeAL (Xiao et al., 2023)) or in-context learning optimization methods (Margatina et al., 2023). A key limitation is that they do not assess the model's mastery of learned and to-be-learned content nor analyze the correlation between the model's latent space distribution and the semantic features of upcoming knowledge. This hinders the controllable training of LLMs and the incremental expansion of its unmastered knowledge.

To address these issues, this paper introduces the Knowledge-Aware Active Learning (KA2L) framework, which pioneers a new paradigm based on semantic consistency. Within this paradigm, we operationally define an LLM's "unknown knowledge" as its inability to stably generate semantically consistent answers to a factual question. This phenomenon is quantified by high Semantic Entropy (SE) (Farquhar et al., 2024; Kuhn et al., 2023), a metric at the heart of our approach. The core objective of this framework is thus to accurately assess this knowledge boundary, thereby efficiently guiding the construction of SFT datasets. Specifically, the KA2L framework employs a Knowledge Distribution Probing mechanism that utilizes hidden states from specific Transformer layers. It performs clustering based on semantic entailment and uses SE to unsupervisedly train a Multi-Layer Perceptron (MLP) as a classifier, which categorizes the question set into "Known" and "Unknown" parts. Furthermore, the KA2L framework utilizes a "hidden-state decoding" technique to "reverse engineer" a large volume of natural-sounding questions from the hidden-space representations corresponding to knowledge points identified within the "Unknown" regions. By incorporating these questions into the training data, the KA2L framework establishes an active learning closed loop, enabling the model to concentrate on learning knowledge it has not yet mastered and thereby minimizing repetitive learning and annotation redundancy.

The main contributions of this paper can be summarized as follows:

1. We propose a novel Knowledge-Aware Active Learning framework (KA2L) that can accurately assess an LLM's degree of knowledge mastery and, by integrating hidden state decoding techniques, actively mine the model's unknown knowledge to guide efficient incremental learning.

2. We innovatively integrate the problem of LLM knowledge distribution probing with the concepts of hallucination detection, proposing methods for probing and decoding based on hidden states, thereby offering new avenues for understanding and shaping the internal knowledge representations of LLMs.

3. Through extensive experiments on nine open-source LLMs and three datasets, we demonstrate that KA2L not only achieves performance comparable to fine-tuning on the full dataset while reducing annotation and computational costs by approximately $50\%$, but also significantly outperforms adapted classic active learning methods, including Coreset and BADGE, providing a novel and cost-effective solution for fine-tuning LLMs.

## 2 RELATED WORK

**Active Learning for LLMs.** Active learning is a well-established field for reducing data annotation costs, with classic strategies primarily pivoting on principles of uncertainty, diversity, or a hybrid of both. Prominent methods include uncertainty sampling (e.g., using prediction entropy), diversity-based approaches like Coreset (Sener & Savarese, 2018) which selects a representative subset of data, and hybrid methods such as BADGE (Ash et al., 2020) that unify both principles via gradient embeddings. However, transplanting these methods, originally designed for models like CNNs, to modern generative LLMs presents significant challenges. For instance, gradient-based methods like BADGE or Fisher information-based methods like BAIT (Ash et al., 2021) become computationally prohibitive due to the immense scale of LLM parameters, and their core logic does not straightforwardly apply to generative tasks. To our knowledge, systematic adaptation and evaluation of these classic methods for LLM fine-tuning has been limited. In our work, we implement practical adaptations of these strategies to compare with our methods. Parallel to this, other LLM-specific active learning research has focused on alternative goals, such as collaborative learning without human supervision (Xiao et al., 2023) or optimizing demonstrations for in-context learning (Margatina et al.,

2023). In contrast to these approaches, our work introduces a fundamentally different selection signal, semantic entropy, derived from the semantic consistency across multiple model generations. This allows us to directly probe the model's knowledge stability, bypassing the computational hurdles of classic methods while offering a more direct proxy for knowledge gaps in generative tasks. See Appendix B for additional related work.

## 3 METHOD

### 3.1 PROBLEM FORMULATION

Given the abstract nature of knowledge, this study considers "questions" as external manifestations of knowledge; i.e., a model's ability to correctly answer a question signifies its possession of the knowledge represented by that question. Given an LLM, denoted as $M$, and a set of questions $Q = \{q_1, q_2, \ldots, q_n\}$, the knowledge distribution of model $M$ over $Q$, denoted $K_{M,Q}$, is defined as a partition of $Q$ into $(Q_k, Q_{unk})$. Here, $Q_k$ represents the set of questions the model can answer correctly with high confidence, while $Q_{unk}$ represents the set of questions for which the model's answers are uncertain. $Q_{unk}$ will guide knowledge mining, as well as dataset annotation and model fine-tuning in downstream tasks.

To mitigate the risk that $Q_{unk}$ may be too small to effectively support downstream fine-tuning, this study further investigates a question augmentation strategy grounded in the model's internal representations. For any question $q_i \in Q_{unk}$, its internal hidden states generated by model $M$ during processing are utilized. A new set of questions $\{q_i^{(1)}, q_i^{(2)}, \ldots, q_i^{(k_i)}\}$ ($k_i \geq 0$), similar in domains and knowledge points, is then generated through hidden state decoding techniques. The augmented set of unknown questions is represented as $Q_{aug} = \bigcup_{q_i \in Q_{unk}} (\{q_i\} \cup \{q_i^{(1)}, \ldots, q_i^{(k_i)}\})$.

Downstream tasks include SFT dataset construction and model fine-tuning. Dataset construction can be defined as creating $\mathcal{D}_{unk} = \{\langle q_i, a_i \rangle | q_i \in Q_{aug}\}$, where $a_i$ is the ground-truth answer to $q_i$. The objective of model fine-tuning is formulated as follows:

$$\theta_{ft} = \arg\min_{\theta} \frac{1}{N} \sum_{(q_i, a_i) \in \mathcal{D}_{unk}} Loss(M(q_i; \theta), a_i) \tag{1}$$

where $\theta_{ft}$ are the fine-tuned model parameters, $\theta$ are the parameters to be optimized during fine-tuning, $N$ is the size of the dataset $\mathcal{D}_{unk}$, $Loss$ is the loss function, and $M$ is the model. Downstream tasks are not the primary focus of this study.

### 3.2 KNOWLEDGE DISTRIBUTION PROBING

The objective of knowledge distribution probing is to identify the intrinsic distribution of "known" and "unknown" questions for an LLM $M$ over a specific question set $Q$. This task is highly correlated with the goal of LLM hallucination detection. As illustrated in Figure 1 (a), our probing framework utilizes semantic entropy (Farquhar et al., 2024; Kuhn et al., 2023) as a metric for quantifying uncertainty in model outputs and employs a Multi-Layer Perceptron (MLP) as the classifier to distinguish whether the knowledge corresponding to the hidden state is mastered by the model. The main workflow includes hidden state and model output sampling, semantic entropy calculation and label construction, classifier training, and inference.

#### 3.2.1 HIDDEN STATE AND MODEL OUTPUT SAMPLING

As shown in Figure 1 (a), for each question $q_i$ in the question set $Q$, the hidden states $H_i = \{h_i^l\}_{l=1}^{L}$ of the last token across all $L$ layers are obtained from LLM $M$ at a low temperature (e.g., temperature=0.1). Subsequently, for the same question $q_i$, multiple independent samples are drawn at a higher decoding temperature, yielding a set of output sentences $S_i = \{s_i^{(1)}, s_i^{(2)}, \ldots, s_i^{(k)}\}$. This process constructs the dataset $\mathcal{S} = \{\langle q_i, H_i, S_i \rangle\}_{q_i \in Q}$ for subsequent classifier training.

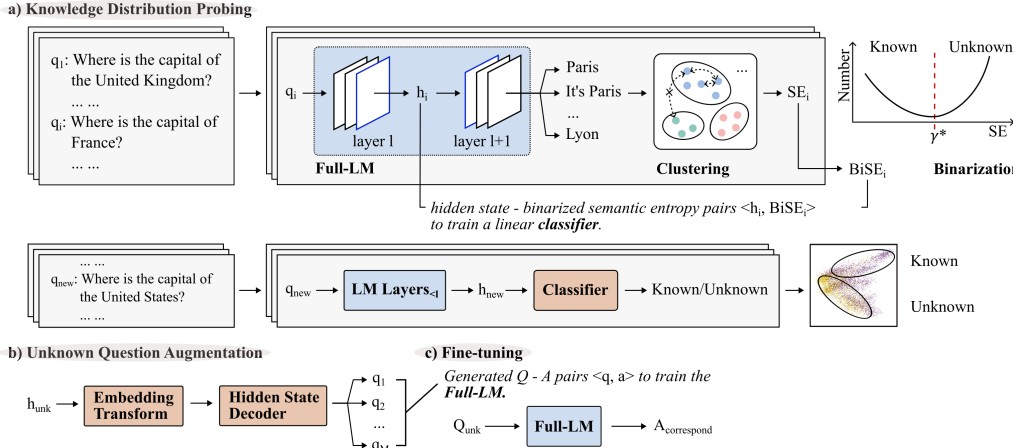

Figure 1: **KA2L Workflow**: **(a) Knowledge Distribution Probing**: **Training Phase**: For each question in the sampled question set, sample its hidden state once and its textual outputs multiple times. Perform semantic clustering on the textual outputs to calculate Semantic Entropy (SE). The SE is then binarized using a dynamic threshold to serve as labels for the classifier. An MLP classifier is trained using these hidden states and the binarized SE (BiSE). **Inference Phase**: For a new set of questions, sample their $l$-th layer hidden states. These are then classified by the MLP classifier, representing the knowledge distribution as "Known" and "Unknown" knowledge. **(b) Unknown Question Augmentation**: Sample the hidden states ($h_{unk}$) of questions identified as "Unknown" from the knowledge distribution. These are then transformed and decoded into multiple similar questions. **(c) Downstream Tasks**: This knowledge distribution guides dataset construction and model fine-tuning. Many existing methods can be applied, such as LoRA (Hu et al., 2022) and P-tuning (Liu et al., 2022).

### 3.2.2 SEMANTIC ENTROPY CALCULATION AND LABEL CONSTRUCTION

The calculation of semantic entropy (SE) follows the methodology proposed in Farquhar et al. (2024); Kuhn et al. (2023). It involves performing semantic clustering on the multiple sampled outputs $S_i$ for the same question $q_i$ and quantifying the consistency of the output content based on the clustering results. A lower SE value indicates higher semantic consistency across multiple outputs; conversely, a higher value suggests greater semantic divergence, indicating that the model has not mastered the knowledge associated with the question. Semantic clustering employs a pre-trained Natural Language Inference (NLI) model (e.g., DeBERTa (He et al., 2021)) to determine semantic equivalence between sentences: if sentence A entails sentence B and sentence B entails sentence A, they are considered semantically equivalent.

For the output set $S_i$ of question $q_i$, let the set of semantic equivalence classes be $\mathcal{C}_{q_i} = \{c_1, c_2, \ldots, c_{|\mathcal{C}_{q_i}|}\}$, where $c_k$ is an equivalence class, $|c_k|$ is the number of sentences in that class, and $N = \sum_k |c_k| = |S_i|$ is the total number of samples. The semantic entropy $\mathrm{SE}(S_i)$ can be estimated as:

$$\mathrm{SE}(S_i) \approx -\sum_{k=1}^{|\mathcal{C}_{q_i}|} \frac{|c_k|}{N} \ln \frac{|c_k|}{N} \tag{2}$$

To obtain binary labels (known/unknown) for classifier training, a dynamic thresholding method is applied to binarize the calculated $\mathrm{SE}(S_i)$. Specifically, let $\mathcal{T} = \{\tau_1, \tau_2, \ldots, \tau_K\}$ be $K$ candidate thresholds selected within the range of all sample SE values $[\min(\{\mathrm{SE}_i\}_{i=1}^{|Q|}), \max(\{\mathrm{SE}_i\}_{i=1}^{|Q|})]$. For any candidate threshold $\tau \in \mathcal{T}$, each sample's $\mathrm{SE}_i$ is binarized, and the mean-square error (MSE) between its binarized result and the original continuous semantic entropy is calculated:

$$\mathrm{MSE}(\tau) = \frac{1}{|Q|} \sum_{i=1}^{|Q|} (\mathrm{SE}_i - \mathbb{I}(\mathrm{SE}_i \geq \tau))^2 \tag{3}$$

where $\mathbb{I}(\cdot)$ is the indicator function, which is 1 if the condition is true and 0 otherwise. Subsequently, the optimal threshold $\gamma^*$ is computed, and $\mathrm{SE}_i$ is binarized to obtain $\mathrm{BiSE}_i$. (See Appendix F.6 for a detailed robustness analysis demonstrating the effectiveness of this dynamic thresholding method).

$$\gamma^* = \arg\min_{\tau \in \mathcal{T}} \mathrm{SE}(\tau) \tag{4}$$

$$\mathrm{BiSE}_i = \begin{cases} 0 & \text{if } SE_i < \gamma^* \quad \text{(representing known)} \\ 1 & \text{if } SE_i \geq \gamma^* \quad \text{(representing unknown)} \end{cases} \tag{5}$$

Finally, the classifier training dataset $\mathcal{D}_{\mathrm{clf}} = \{\langle H_i, \mathrm{BiSE}_i\rangle\}_{q_i \in Q}$ is constructed.

### 3.2.3 CLASSIFIER TRAINING AND INFERENCE

The dataset $\mathcal{D}_{\mathrm{clf}}$ is partitioned into training, validation, and test sets in a 7:2:1 ratio. To effectively discriminate the knowledge states of the LLM, we design a Multi-Layer Perceptron (MLP) as the hidden state classifier. The architectural design of this MLP is informed by efficient MLP components found in modern large language models, such as Llama3, comprising 4 linear layers and 1 SiLU activation function. Considering the high dimensionality and potentially complex non-linear features of LLM hidden states, we opted for an MLP structure, aiming for stronger representation learning and pattern recognition capabilities compared to traditional linear classifiers (e.g., logistic regression). The classifier's input dimension matches the hidden state dimension $h_i^l$ of the LLM $M$ under test, the output dimension is 2, and the intermediate layer dimension is set to 14336. Training utilizes the standard CrossEntropyLoss function and Adam optimizer (Kingma & Ba, 2017), with a learning rate of $1.0e-5$ for 20 epochs. For all $L$ hidden layers of each LLM $M$, $L$ independent classifiers are trained. The best-performing classifier $C$ and its corresponding hidden layer number $l$ are selected based on their performance on the test set.

For each question in a new question set $Q_{new}$, we extract the hidden state of the last token at the selected layer $l$. This hidden state is then fed into the trained classifier $C$ for classification, yielding a determination of whether the question is "known" or "unknown," ultimately constructing the knowledge distribution over $Q_{new}$. The classifier $C$ is characterized by its fast operational speed and high parallelizability, rendering its performance overhead on the overall active learning framework negligible. Questions identified as falling within the "Unknown" region of the knowledge distribution are collected and subsequently directed to the hidden state decoding process.

### 3.3 UNKNOWN QUESTION AUGMENTATION VIA HIDDEN STATE DECODING

To augment the "Unknown" question set, we employ a latent space decoding technique from LLM interpretability research. This approach transcends surface-level paraphrasing by decoding the model's rich, abstract hidden states (Morris et al., 2024; Geva et al., 2021) into new, diverse questions. These generated questions probe the same knowledge points from different perspectives, thereby more comprehensively identifying the model's knowledge deficiencies. We adapt the "vec2text" method (Morris et al., 2024) by training a dedicated decoder to translate hidden states into natural language text, as detailed in Appendix E.1.

## 4 EXPERIMENTAL SETUP

To rigorously evaluate the effectiveness and efficiency of our proposed KA2L framework, we conducted a series of experiments.

### 4.1 MODELS, DATASETS, AND EVALUATION METRICS

The experiments selected nine open-source large language models, including Llama (Touvron et al., 2023), Mistral (Jiang et al., 2023), Phi (Abdin et al., 2024), Qwen (Qwen et al., 2025), and GLM (GLM et al., 2024), covering different model architectures and parameter scales. To ensure the breadth of the evaluation, the experiments utilized 2 open-domain question-answering datasets, TriviaQA (Joshi et al., 2017) and NQ_Open (Lee et al., 2019), and 1 medical domain dataset, MedMCQA (Pal et al., 2022). These datasets cover different knowledge domains and question types.

Evaluation metrics including BLEU (Papineni et al., 2002), ROUGE (Lin, 2004), METEOR (Lavie & Agarwal, 2007), and BERTScore (Zhang* et al., 2020), were adopted to comprehensively assess the fluency, accuracy, and semantic similarity of the generated answers.

It is important to note that all our experiments are conducted in a closed-book setting. This means the models rely solely on their internal, parametric knowledge to answer questions, without access to any external information retrieval system during inference. This setup is crucial for our goal, which is to evaluate the effectiveness of directly injecting knowledge into the model's parameters through targeted fine-tuning.

## 4.2 VALIDATING KA2L'S EFFICACY

The core of our experimental design is to validate the efficacy of KA2L through three central research questions (RQs). The process begins by training a knowledge distribution probe for each LLM-dataset pair on a held-out data sample. This trained probe then partitions a separate, larger data pool to construct our primary fine-tuning datasets:

$D_{\mathbf{unk}}$ Contains questions identified by the probe as "unknown". This set represents the high-value data actively selected by our KA2L framework.

$D_{\mathbf{k}}$ Contains questions identified as "known", serving as a baseline to evaluate the utility of data already mastered by the model.

$D_{\mathbf{combine}}$ A balanced mix of samples from $D_{\mathrm{unk}}$ and $D_{\mathrm{k}}$, simulating a standard, unfiltered dataset collected without an active learning strategy.

For each model-dataset pair, the trained probe is used to select $10,000$ "unknown" and $5,000$ "known" samples from the larger data pool, forming the basis for our experimental sets. The probe extracts hidden states from the layer which yield the highest classification accuracy (see Appendix F.4 for analysis and Table 4 for final layer selections). Specifically, the $10k$ *Unknown* dataset consists of all $10,000$ selected unknown samples. The $5k$ *Unknown* dataset is a random $5,000$-sample subset of this $10k$ set. The $10k$ *Combine* dataset is constructed by mixing $5,000$ of the unknown samples with the $5,000$ selected known samples.

We fine-tune each model using LoRA with standard configurations (see Appendix E.2 for a comprehensive list of all fine-tuning hyperparameters).

**RQ1: Cost-Efficiency.** Can KA2L achieve comparable performance to a full, unfiltered dataset while using only a fraction (e.g., $50\%$) of the annotation and computational budget? To answer this, we compare fine-tuning on *5k Unknown* data (a $5,000$-sample subset of $D_{\mathrm{unk}}$) against $10k$ *Combine* data (from $D_{\mathrm{combine}}$). This tests if our method can match a larger dataset's performance with half the budget.

**RQ2: Selection Effectiveness.** Given an identical data budget, does KA2L's strategy of selecting "unknown" data yield superior performance compared to a naive, unfiltered approach? Here, we compare fine-tuning on $10k$ *Unknown* data against the same $10k$ *Combine* baseline. This directly isolates the benefit of focusing on knowledge gaps versus indiscriminate training.

**RQ3: Augmentation Utility.** In scenarios with a limited pool of original "unknown" data, can our hidden-state decoding method effectively augment the training set to further boost performance? To investigate this, we create a $10k$ *Augmented* dataset by generating $5,000$ new questions from the hidden states of the $5k$ *Unknown* set. We then compare the fine-tuning performance of this augmented set against the original $5k$ *Unknown* set (to measure the uplift from augmentation) and the $10k$ *Unknown* set (to gauge how closely synthetic data can approximate additional original data).

## 4.3 SUPPLEMENTARY ANALYSES

To provide deeper insights into our framework and validate its components, we conducted several supplementary analyses. Detailed methodologies and results are presented in the Appendix.

**Comparison with Traditional AL.** We compared KA2L against adapted traditional active learning methods to assess its effectiveness. The detailed setup and full results are discussed in §5.3, Appendix E.3 and F.2.

**Component Validation.** We assessed the performance of our Knowledge Distribution Probe (§5.2, Appendix F.3) and investigated the robustness of our Dynamic Thresholding method (Appendix F.6).

Table 1: **Active learning performance on MedMCQA.** KA2L-selected data (Unknown) is compared against Known data and a mixed Combine setting. Full results for other datasets are in the Appendix F.1.

| Model | SFT Dataset | BLEU | ROUGE-L | METEOR | BS(%) | SFT Dataset | BLEU | ROUGE-L | METEOR | BS(%) |
|---|---|---|---|---|---|---|---|---|---|---|
| DeepSeek-R1-Distill-Qwen-7B | None | 0.02 | 0.52 | 1.31 | 76.29 | 10k Combine | 2.44 | 12.55 | 7.92 | 83.70 |
| | 5k Known | 1.85 | 10.41 | 6.50 | 83.19 | 10k Unknown | **2.85** | **14.53** | **9.20** | **84.00** |
| | 5k Unknown | 1.91 | 11.99 | 7.54 | 83.62 | 10k Augmented | 2.62 | 13.55 | 8.91 | **84.00** |
| glm4-9b-chat | None | 0.08 | 1.71 | 4.21 | 78.48 | 10k Combine | 6.90 | 28.27 | 19.67 | 86.80 |
| | 5k Known | 4.17 | 20.62 | 13.86 | 85.17 | 10k Unknown | **9.82** | **36.02** | **25.50** | **88.30** |
| | 5k Unknown | 6.61 | 28.44 | 19.87 | 86.73 | 10k Augmented | 8.48 | 29.92 | 21.67 | 87.13 |
| Llama-2-7b-chat-hf | None | 0.06 | 0.99 | 2.54 | 77.33 | 10k Combine | 5.37 | 23.76 | 15.80 | 85.90 |
| | 5k Known | 3.34 | 16.13 | 10.45 | 84.35 | 10k Unknown | **7.99** | **29.78** | **20.14** | **87.15** |
| | 5k Unknown | 5.91 | 23.25 | 15.54 | 85.79 | 10k Augmented | 5.56 | 23.77 | 16.06 | 85.92 |
| Llama-3.1-8B-Instruct | None | 0.10 | 2.34 | 5.01 | 78.58 | 10k Combine | 8.21 | 30.17 | 21.11 | 87.14 |
| | 5k Known | 5.72 | 23.29 | 16.18 | 85.71 | 10k Unknown | **10.82** | **36.55** | **25.97** | **88.49** |
| | 5k Unknown | 8.49 | 29.96 | 20.84 | 87.20 | 10k Augmented | 8.81 | 30.54 | 21.36 | 87.31 |
| Mistral-7B-Instruct-v0.1 | None | 0.12 | 3.17 | 6.29 | 79.60 | 10k Combine | 7.06 | 27.86 | 19.16 | 86.70 |
| | 5k Known | 3.35 | 18.09 | 12.03 | 84.74 | 10k Unknown | **9.71** | **36.11** | **25.20** | **88.37** |
| | 5k Unknown | 6.97 | 27.33 | 18.82 | 86.65 | 10k Augmented | 7.43 | 28.85 | 20.29 | 86.92 |
| Mistral-7B-Instruct-v0.3 | None | 0.11 | 2.10 | 5.19 | 79.22 | 10k Combine | 6.88 | 27.55 | 19.29 | 86.76 |
| | 5k Known | 4.32 | 19.17 | 13.11 | 85.03 | 10k Unknown | **9.58** | **35.28** | **25.08** | **88.30** |
| | 5k Unknown | 6.71 | 27.02 | 18.79 | 86.63 | 10k Augmented | 8.24 | 28.94 | 20.86 | 87.07 |
| Phi-3.5-mini-instruct | None | 0.09 | 1.59 | 4.10 | 78.44 | 10k Combine | 7.42 | 27.70 | 18.57 | 86.74 |
| | 5k Known | 6.63 | 25.40 | 16.90 | 86.30 | 10k Unknown | 8.05 | **29.69** | 19.95 | **87.21** |
| | 5k Unknown | 7.61 | 27.85 | 18.59 | 86.81 | 10k Augmented | **8.30** | 29.18 | **20.05** | 87.07 |
| Qwen1.5-7B-Chat | None | 0.08 | 1.64 | 4.03 | 78.63 | 10k Combine | 4.60 | 19.65 | 13.12 | 85.00 |
| | 5k Known | 3.11 | 15.09 | 9.95 | 84.10 | 10k Unknown | **5.46** | **23.72** | **15.78** | **85.84** |
| | 5k Unknown | 4.13 | 19.34 | 12.86 | 84.90 | 10k Augmented | 4.85 | 21.55 | 14.84 | 85.40 |
| Qwen2.5-7B-Instruct | None | 0.09 | 1.80 | 4.34 | 78.35 | 10k Combine | 6.30 | 24.13 | 16.80 | 85.93 |
| | 5k Known | 4.67 | 20.30 | 13.85 | 85.06 | 10k Unknown | 6.63 | **27.16** | **18.79** | **86.54** |
| | 5k Unknown | **6.80** | 24.07 | 16.58 | 85.95 | 10k Augmented | 5.92 | 24.65 | 17.73 | 86.14 |

**Layer-wise Analysis.** We examined how knowledge uncertainty is distributed across different transformer layers for each model (Appendix F.4).

**Data Scaling Effect.** We explored the impact of varying the quantity of "unknown" data on fine-tuning performance (Appendix F.5).

**Qualitative Analysis.** To provide an intuitive validation for our framework's core premise, we present a qualitative case study. This analysis visually demonstrates the strong correlation between a model's output consistency and its underlying knowledge state, thereby supporting our operational definition of "unknown knowledge" as an LLM's inability to stably generate semantically consistent answers to a factual question (Appendix F.7).

## 5 RESULTS AND ANALYSIS

### 5.1 KA2L-GUIDED FINE-TUNING ACHIEVES SUPERIOR COST-EFFICIENCY AND EFFECTIVENESS

Our primary experiments, summarized in Table 1 for the MedMCQA dataset, demonstrate the substantial advantages of the KA2L framework across a diverse set of nine LLMs. These findings are consistently replicated on the open-domain TriviaQA and NQ_Open datasets, as detailed in Appendix F.1. The results provide clear and consistent answers to our core research questions regarding cost-efficiency (RQ1), selection effectiveness (RQ2), and augmentation utility (RQ3).

**Cost-Efficiency (RQ1).** A central finding is that fine-tuning with $5k$ *Unknown* samples, actively selected by KA2L, achieves performance comparable to the $10k$ *Combine* setting while using only half the data. For instance, on the challenging MedMCQA dataset, `Llama-3.1-8B-Instruct` trained on $5k$ *Unknown* data reaches a ROUGE-L score of 29.96, nearly identical to the 30.17 achieved with the full *10k Combine* set. This pattern holds across most models, such as `glm4-9b-chat` ($5k$ *Unknown*: 28.44 vs. $10k$ *Combine*: 28.27). This directly validates that our framework can cut annotation and computational costs by approximately $50\%$ while maintaining high performance, confirming the cost-efficiency of focusing on unmastered knowledge.

Table 2: **Performance comparison of knowledge distribution probes (AUROC) on TriviaQA dataset.** Our method achieves the highest score. Similar trends are observed on MedMCQA and NQ_Open (Appendix F.3)

| Models | Ours | SE Probe | Accuracy Probe | Log-Likeli hood | Regular Entropy | P(True) | Semantic Entropy |
|---|---|---|---|---|---|---|---|
| DeepSeek-R1-Distill-Qwen-7B | **0.89** | 0.85 | 0.81 | 0.61 | 0.84 | 0.83 | 0.86 |
| glm4-9b-chat | **0.83** | 0.81 | 0.72 | 0.56 | 0.76 | 0.82 | 0.80 |
| Llama-2-7b-chat-hf | **0.85** | 0.78 | 0.70 | 0.55 | 0.73 | 0.71 | 0.77 |
| Llama-3.1-8B-Instruct | **0.89** | 0.85 | 0.73 | 0.57 | 0.75 | 0.79 | 0.79 |
| Mistral-7B-Instruct-v0.1 | **0.88** | 0.83 | 0.76 | 0.62 | 0.75 | 0.78 | 0.81 |
| Mistral-7B-Instruct-v0.3 | **0.90** | 0.86 | 0.78 | 0.68 | 0.73 | 0.84 | 0.78 |
| Phi-3.5-mini-instruct | **0.91** | 0.88 | 0.81 | 0.77 | 0.82 | 0.79 | 0.84 |
| Qwen1.5-7B-Chat | **0.81** | 0.78 | 0.74 | 0.46 | 0.77 | 0.78 | **0.81** |
| Qwen2.5-7B-Instruct | **0.86** | 0.82 | 0.75 | 0.54 | 0.79 | **0.86** | 0.80 |

**Effectiveness of Selection (RQ2).** When comparing datasets of the same size, the superiority of KA2L becomes even more apparent. The $10k$ *Unknown* setting consistently and significantly outperforms the $10k$ *Combine* across all models and metrics. For example, `Llama-3.1-8B-Instruct` achieves a ROUGE-L of 36.55 with $10k$ *Unknown* data, a remarkable 6.38-point improvement over the 30.17 from the $10k$ *Combine* set. Similar substantial gains are observed for all other models, such as `Mistral-7B-Instruct-v0.3` (35.28 vs. 27.55). This result powerfully illustrates that given a fixed budget, intelligently selecting the model's unknown data is far more effective than an unfiltered training approach. It also highlights that fine-tuning on "known" data, as is done in $10k$ *Combine* setting, offers limited benefits and can be considered redundant.

**Utility of Data Augmentation (RQ3).** Our experiments with the $10k$ *Augmented* set reveal its practical utility in data-scarce scenarios. As expected, the performance of augmented data generally does not reach the ceiling set by an equivalent amount of original $10k$ *Unknown* data, since original samples contain the most novel information. However, the augmented set consistently provides a significant performance boost over the $5k$ *Unknown* set. For instance, with `Phi-3.5-mini`, the $10k$ *Augmented* set (ROUGE-L 29.18) notably improves upon the $5k$ *Unknown* set (27.85) and even slightly outperforms the unfiltered $10k$ *Combine* set (27.70). This result highlights that when acquiring more original "unknown" data is costly or infeasible, our hidden-state decoding method offers a practical and effective way to enrich the training data and further improve model performance.

## 5.2 Validating the Knowledge Distribution Probe

The efficacy of our entire KA2L framework hinges on the performance of its core component: the knowledge distribution probe. To validate its ability to accurately identify a model's knowledge gaps, we evaluate it on the task of classifying questions as "known" or "unknown". We benchmark our MLP-based probe against a suite of strong baselines from hallucination detection and uncertainty quantification literature, using AUROC as the primary metric.

As demonstrated in Table 2, our probe consistently achieves state-of-the-art performance across all nine evaluated LLMs on the TriviaQA dataset. Notably, it attains an AUROC of up to 0.91 (on `Phi-3.5-mini-instruct`), establishing a new performance benchmark for this task. It significantly outperforms SE Probe, the most conceptually similar baseline, which also leverages hidden states but relies on a simpler logistic regression classifier. This underscores the effectiveness of our MLP architecture in capturing the complex, non-linear uncertainty signals encoded within the transformer's internal representations.

Furthermore, compared to methods that operate on model outputs, our approach shows a distinct advantage. It surpasses both surface-feature-based methods like Semantic Entropy (SE) and logit-based methods such as Log-Likelihood and Regular Entropy. This suggests that the internal hidden states provide a more reliable and direct signal of a model's epistemic uncertainty than its final output distribution or lexical variations. Crucially, this high accuracy is achieved with a lightweight MLP classifier, ensuring that the probe introduces negligible computational overhead during inference. The probe's superior and efficient performance provides a robust foundation for the KA2L framework, ensuring that the data selected for active learning is of the highest value.

Table 3: **Performance comparison of active learning methods on the NQ_Open dataset.** Our method, KA2L ($5k$ **Unknown**), significantly outperforms all adapted traditional AL method, approaching the performance of the **Full Dataset** ($10k$) with only half the data budget. Results are reported as mean ± std over 4 runs.

| Method | BLEU | ROUGE-L | METEOR | BertScore(%) |
|---|---|---|---|---|
| *Traditional Methods (5k samples selected from 10k pool)* | | | | |
| Random | $21.75 \pm 0.03$ | $38.53 \pm 0.02$ | $29.65 \pm 0.02$ | $90.01 \pm 0.01$ |
| Entropy | $19.59 \pm 1.15$ | $37.91 \pm 0.10$ | $29.38 \pm 0.07$ | $89.91 \pm 0.01$ |
| Coreset | $22.04 \pm 0.07$ | $38.69 \pm 0.05$ | $29.81 \pm 0.03$ | $90.16 \pm 0.01$ |
| BADGE (adapted) | $19.64 \pm 0.15$ | $38.70 \pm 0.05$ | $29.73 \pm 0.05$ | $90.06 \pm 0.01$ |
| *Our Method (5k samples)* | | | | |
| KA2L 5k Known | $16.68 \pm 0.09$ | $32.58 \pm 0.03$ | $24.68 \pm 0.03$ | $89.07 \pm 0.01$ |
| **KA2L 5k Unknown** | $\mathbf{22.29 \pm 0.08}$ | $\mathbf{44.96 \pm 0.04}$ | $\mathbf{35.01 \pm 0.04}$ | $\mathbf{91.08 \pm 0.01}$ |
| *Upper Bound* | | | | |
| Full Dataset (10k) | $24.51 \pm 0.04$ | $45.18 \pm 0.04$ | $35.35 \pm 0.04$ | $91.09 \pm 0.01$ |

### 5.3 COMPAIRSON WITH ADAPTED TRADITIONAL ACTIVE LEARNING METHODS

To situate KA2L in the broader research context, we performed a comparative analysis against several traditional active learning methods. Since methods such as Entropy (Wang & Shang, 2014), Coreset (Sener & Savarese, 2018), and BADGE (Ash et al., 2020) were not originally designed for generative LLMs, it is necessary to develop practical adaptations for them, primarily by using prediction entropy as a proxy for uncertainty and final-layer hidden states as embeddings for diversity. Our full adaptation methodology for these strategies is detailed in Appendix E.3. This comparative study, conducted on the `LLaMA-3.1-8B-Instruct` model, investigates how KA2L's LLM-native selection signal performs relative to these adapted strategies. As shown in Table 3, the results highlight a significant performance gap.

On the NQ_Open dataset, fine-tuning on the 5k Unknown set selected by KA2L achieves a ROUGE-L score of $44.96 \pm 0.04$. This not only significantly surpasses all traditional AL methods but is also remarkably close to the performance of the Full Dataset ($45.18 \pm 0.04$), which uses twice the amount of data. Full results on all three datasets, presented in Appendix F.2 (Tables 8 and 9), consistently corroborate this finding, demonstrating that KA2L's semantic-uncertainty-based approach is more effective at identifying informative samples for knowledge-intensive tasks than strategies based on general uncertainty, diversity, or adapted gradients.

## 6 CONCLUSION

In this paper, we introduced the Knowledge-Aware Active Learning (KA2L) framework, a novel approach for efficiently fine-tuning Large Language Models. By probing the model's internal hidden states to identify "unknown" knowledge, KA2L guides a more targeted and cost-effective data selection process. Our extensive experiments demonstrate that fine-tuning with KA2L-selected data not only reduces annotation and computation costs by approximately $50\%$ but also achieves superior performance compared to both unfiltered datasets and classic active learning methods like Coreset and BADGE. The core of our framework, a highly accurate knowledge probe, effectively pinpoints a model's knowledge boundaries, while our hidden-state decoding offers a practical solution for data augmentation in low-resource scenarios. Ultimately, KA2L presents a practical and robust solution for targeted LLM enhancement, highlighting the significant value of leveraging a model's internal knowledge distribution for more efficient learning. The limitations of our current framework and promising future directions are detailed in Appendix C and D.

## 7 REPRODUCIBILITY STATEMENT

To ensure reproducibility, our full source code is publicly available in an anonymous repository at `https://anonymous.4open.science/r/KA2L-F15C`. The Appendix provides comprehensive implementation details, including all hyperparameters (Appendix E) and additional results (Appendix F), to facilitate the complete replication of our findings.

## 8 ETHICS STATEMENT

Our research exclusively utilizes publicly available and widely used benchmark datasets (TriviaQA, NQ_Open, and MedMCQA). No new data was collected from human subjects. Our work aims to improve the efficiency of knowledge acquisition in LLMs, a fundamentally positive scientific goal. We have reviewed the ICLR Code of Ethics and, to the best of our knowledge, our methodology does not raise any direct negative societal impacts or ethical concerns.

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

## A    LLM Usage Statement

During the preparation of this manuscript, we utilized LLMs for assistance with language translation and polishing to improve grammatical clarity. All core research ideas, experimental design, data analysis, and the final conclusions were conceived and formulated exclusively by the authors. The authors take full responsibility for all content presented in the paper.

## B    Additional Related Work

**LLM Hallucination Detection.** Given that hallucinations (Shuster et al., 2021) are a direct manifestation of an LLM's unknown knowledge, we define the problem of identifying LLM knowledge distribution as a hallucination detection task, also termed output uncertainty quantification (Li et al., 2023). Current methods fall into two main categories. Output-based methods assess uncertainty from surface features like text or logits. Examples include sampling-based consistency checks (Manakul et al., 2023) and Semantic Entropy (SE) (Farquhar et al., 2024; Kuhn et al., 2023), which quantifies semantic similarity across multiple outputs to handle linguistic variations. However, these methods often incur high computational costs due to repeated sampling (Kossen et al., 2024). In contrast, probing methods leverage internal hidden states, training a classifier to predict output uncertainty (Azaria & Mitchell, 2023; Li et al., 2023). Azaria & Mitchell (2023) demonstrated that hidden states contain veracity signals, enabling efficient detection. More recently, Semantic Entropy Probes (SEP) (Kossen et al., 2024) were proposed, training a classifier to predict SE values directly from hidden states. By using SE as an unsupervised learning target, SEP achieves performance comparable to surface SE methods but with significantly lower computational cost and improved generalization.

**LLM Hidden State Decoding.** One of the core ideas of this research is to leverage LLM hidden state decoding techniques to mine potential related knowledge from the internal representations generated when the model processes specific questions (particularly those in the "Unknown" region)(Lv et al., 2024; Tang et al., 2025). This allows for the generation of new questions that are diverse in form yet related in terms of knowledge points, effectively augmenting the set of unknown questions. To elucidate the foundation and existing advancements of the techniques adopted in this study, this section will review relevant hidden state decoding methods. nostalgebraist (2020); Sakarvadia et al. (2023); Belrose et al. (2023); Pal et al. (2023), based on "lens" methods, proposed an approach for directly decoding hidden states by applying minor transformations and utilizing the model's pre-trained unembedding module to convert hidden states directly into logits. The Patchscope (Ghandeharioun et al., 2024) and SelfIE (Chen et al., 2024) methods patch hidden states into another model, whose output serves as the decoded result, offering higher readability than lens methods. Morris et al. (2023; 2024) introduced the Vec2Text method, which trains a T5 (Raffel et al., 2020) model as a decoder to transform hidden state vectors into natural language text. Experiments demonstrated high accuracy and readability when decoding later-layer hidden state vectors from the LLaMA2 model.

## C    Limitations

While our KA2L framework effectively identifies questions representing unmastered knowledge to guide SFT dataset construction, its current functionality has several limitations. Firstly, the framework's scope is confined to question classification and prioritization; it does not extend to the automatic generation of complete question-answer pairs for the identified unknown questions, a step that still necessitates external mechanisms or human annotation. Secondly, as a white-box approach, KA2L requires access to the model's internal hidden states, rendering it incompatible with commercial, closed-source models accessible only via APIs. Lastly, the initiation of the KA2L framework is contingent upon a pre-existing, domain-specific question set to probe the LLM's knowledge distribution. The collection and curation of this initial, comprehensive corpus can pose a significant practical challenge, particularly in niche or emerging domains where such resources are scarce, thereby potentially limiting the immediate applicability or bootstrapping of our method in these scenarios.

Table 4: Model-specific layer index choices.

| Model | Hidden-State Layer Index | | |
|-------|-------------|-----------|-------------|
| | TriviaQA Mix | NQ-Open Mix | MedMCQA Mix |
| DeepSeek-R1-Distill-Qwen-7B | 28 | 28 | 28 |
| glm4-9b-chat | 38 | 38 | 24 |
| Llama-2-7b-chat-hf | 31 | 30 | 30 |
| Llama-3.1-8B-Instruct | 31 | 32 | 31 |
| Mistral-7B-Instruct-v0.1 | 31 | 31 | 30 |
| Mistral-7B-Instruct-v0.3 | 31 | 31 | 31 |
| Phi-3.5-mini-instruct | 32 | 32 | 32 |
| Qwen1.5-7B-Chat | 32 | 26 | 19 |
| Qwen2.5-7B-Instruct | 22 | 19 | 28 |

## D  FUTURE WORKS

Our work leads to two main future research directions. A key challenge that warrants further investigation is the inherent difficulty for semantic consistency-based methods, including Semantic Entropy, to fully disambiguate between genuine knowledge gaps (i.e., the model truly does not know) and responses to questions with intrinsic ambiguity or controversy. Although our current study mitigates this issue by focusing on factual question-answering datasets, addressing this distinction in more general and open-ended scenarios remains a significant open problem. Another promising direction involves moving beyond the empirical selection of the optimal layer for uncertainty quantification. Future work could delve into the information flow mechanisms within diverse LLM architectures to develop a more principled understanding of why specific layers are more sensitive to representing knowledge uncertainty. This line of inquiry not only promises to enhance the robustness and theoretical grounding of our framework but also contributes valuable insights to the broader field of model interpretability.

## E  EXPERIMENTAL DETAILS

### E.1  HIDDEN-STATE DECODER TRAINING DETAILS

**Dataset**: This paper used $250k$ data entries for training. Among these, $200k$ entries are fixed, sampled from one-million-instructions (Morris et al., 2024), to ensure its capability in decoding fundamental questions. An additional $50k$ entries originate from the TriviaQA (Joshi et al., 2017), NQ_Open (Lee et al., 2019), and MedMCQA (Pal et al., 2022) datasets used in the experiments of this paper. This subset of data was not used in any other experiments and was solely dedicated to training the decoder for each model on its corresponding dataset.

**Training Parameters**: All experiments utilized a fully fine-tuned t5-base[1] (Raffel et al., 2020) model as the decoder. The training involved 40 epochs, a learning rate of $2.0e - 4$, and $100,000$ warmup steps. The dataset for training the decoder was selected based on the model being decoded and the specific dataset context. The layer numbers for hidden state extraction, presented in Table 4, were not chosen arbitrarily. Instead, they were empirically determined for each model-dataset pair by selecting the layer that yielded the highest classifier performance (AUROC) in our comprehensive layer-wise analysis (see Appendix F.4).

**Decoding Process**: The decoding process leverages the trained t5-base to generate new questions. Specifically, a hidden state $h_{\text{unk}}$ from the "Unknown" region is first transformed into the t5's latent space via two linear layers and a GeLU activation function, with Gaussian noise added to promote diversity. The resulting vector is then fed into the trained t5-base model, which generates the new question in natural language.

---

[1]https://hf-mirror.com/google-t5/t5-base

Table 5: SFT training parameters for different models.

| Model | Learning Rate | Template | Batch Size | Gradient Accumulation Steps |
|---|---|---|---|---|
| DeepSeek-R1-Distill-Qwen-7B | 5.0e-5 | qwen | 4 | 4 |
| glm4-9b-chat | 1.0e-4 | glm4 | 2 | 8 |
| Llama-2-7b-chat-hf | 1.0e-4 | llama2 | 4 | 2 |
| Llama-3.1-8B-Instruct | 1.0e-4 | llama3 | 4 | 2 |
| Mistral-7B-Instruct-v0.1 | 1.0e-4 | mistral | 4 | 4 |
| Mistral-7B-Instruct-v0.3 | 1.0e-4 | mistral | 4 | 4 |
| Phi-3.5-mini-instruct | 1.0e-4 | phi | 4 | 8 |
| Qwen1.5-7B-Chat | 5.0e-5 | qwen | 4 | 4 |
| Qwen2.5-7B-Instruct | 5.0e-5 | qwen | 4 | 4 |

## E.2 FINE-TUNING PARAMETERS

This paper employed the LLaMA-Factory framework (Zheng et al., 2024) to fine-tune all models. The fine-tuning parameters adopted recommended values to simulate real-world fine-tuning scenarios. A total of 9 models were fine-tuned 5 times each across 3 datasets, resulting in 135 fine-tuning runs. For a given model, the only parameter difference when fine-tuning on different datasets was the dataset itself; all other parameters remained consistent.

All fine-tuning was performed using LoRA (Hu et al., 2022), with lora_target set to "all" for all modules. FlashAttention-2 (Dao, 2024), fast_tokenizer, and bf16 were used to accelerate the fine-tuning process. The number of epochs was set to 3 for all runs. Other parameters are detailed in Table 5 below.

For generating outputs from the fine-tuned models during the evaluation phase, a consistent decoding strategy was employed. We used a low temperature of 0.1 for sampling. This encourages the model to produce more factual and deterministic outputs by reducing randomness, which is appropriate for the question-answering tasks in our evaluation. Other decoding parameters, such as top-p and top-k, were kept at their default values.

## E.3 EXPERIMENTAL DETAILS FOR TRADITIONAL ACTIVE LEARNING METHODS

To provide a comparative context for our KA2L framework, we implemented adaptations of several traditional active learning methods. The original design of these methods for classification tasks makes their direct application to generative LLMs computationally intractable or conceptually mismatched. Therefore, our goal was to create practical and faithful adaptations that could serve as meaningful points of comparison. This comparative experiment was performed on the LLaMA-3.1-8B-Instruct model across all three datasets, where each method was tasked with selecting a $5,000$-sample subset from the $10k$ *Combine* set.

Our adaptation strategy centers on a single, efficient pass over the unlabeled data pool to pre-calculate two key metrics for each prompt: an uncertainty score and a diversity embedding. This pre-computation, while resource-intensive, is performed only once, and its results are reused across all comparative methods, ensuring both efficiency and a fair comparison.

**Uncertainty Score.** Traditional gradient-based uncertainty metrics are infeasible for LLMs. We instead use prediction entropy as a computationally efficient and effective proxy. Specifically, for each prompt, we perform a standard forward pass (temperature=1.0) to obtain the next-token logits and calculate the entropy of the resulting probability distribution, $H(p) = -\sum_i p_i \log p_i$. This captures the model's intrinsic confidence and serves as the uncertainty score.

**Diversity Embedding.** To measure diversity, we leverage the semantic representation of each prompt from the LLM itself. Instead of computationally prohibitive gradient embeddings (as in the original BADGE), we extract the hidden-state representation of the final prompt token from a deep model layer (see Table 4). This high-dimensional vector captures the prompt's semantic essence, enabling a meaningful measure of diversity.

Using these two pre-computed metrics, we implement the comparative strategies as follows:

**Random Sampling**: Uniformly samples a subset of data from the pool.

**Uncertainty Sampling**: Selects samples with the highest prediction entropy scores.

**Coreset Sampling**: This diversity-focused method is implemented using a standard k-Center-Greedy algorithm on the hidden-state embeddings, iteratively selecting the data point furthest from its nearest neighbor in the already selected set.

**BADGE (Adapted)**: Our adaptation preserves BADGE's hybrid principle by using a weighted k-MEANS++ seeding procedure. The probability of selecting a new sample is proportional to its squared distance to the nearest selected center, multiplied by its uncertainty score (entropy), balancing diversity and uncertainty.

### E.4 COMPUTATIONAL REQUIREMENTS

All experiments in this paper were conducted on Nvidia A100 40G GPUs. All GPU hours mentioned below are based on the usage of this GPU model.

To provide a practical perspective for practitioners, we first outline the cost for a single model-dataset pair. The process includes:

- **Probe Training and Inference:** Training the knowledge distribution probe on $10,000$ samples and performing inference requires approximately 5 A100-GPU hours. The inference time of the trained probe is negligible.
- **Decoder Training:** Training the T5-based hidden-state decoder on roughly $250k$ samples takes about 20 A100-GPU hours. Its inference is also highly efficient and parallelizable.

It is crucial to interpret these figures as a one-time, upfront investment for a given use case. This initial cost yields a significant return by saving approximately $50\%$ in subsequent annotation and fine-tuning costs, which are often the most expensive and time-consuming parts of the LLM development cycle. This demonstrates the overall cost-effectiveness of our framework.

The total computational budget for the experiments in this paper was approximately 900 GPU hours. This figure reflects the cost of a comprehensive validation process designed to test our framework's robustness and generalizability, rather than the cost of a single, practical application. This large-scale effort, spanning 9 LLMs and 3 datasets, can be broken down as follows:

- A total of 351 GPU hours were dedicated to evaluating the active learning process. This involved training 27 distinct knowledge probes, followed by 135 fine-tuning runs and 162 evaluation runs to compare different data selection strategies.
- A total of 540 GPU hours were allocated to validate the data augmentation component. The majority of this time was spent training the 27 hidden-state decoders required for our analysis.

## F ADDITIONAL EXPERIMENT RESULTS

### F.1 FINE-TUNING PROCESS WITHIN THE ACTIVE LEARNING FRAMEWORK

The experimental results on the open-domain NQ_Open (Table 6) and TriviaQA (Table 7) datasets consistently reinforce the primary findings observed on the MedMCQA dataset. Across all models, every fine-tuning strategy yields substantial improvements over the base models ("None"), confirming the universal benefit of supervised fine-tuning for domain adaptation. The analysis below examines these results through the lens of our core research questions.

**Cost-Efficiency (RQ1).** The cost-efficiency of KA2L is clearly demonstrated, as fine-tuning on $5k$ *Unknown* data consistently achieves performance comparable to the $10k$ *Combine* baseline while using only half the data. For instance, on NQ_Open, `glm4-9b-chat` trained on $5k$ *Unknown* data achieves a ROUGE-L of 45.91, nearly identical to the 45.94 from the $10k$ *Combine* set. Similarly, for `Qwen2.5-7B-Instruct`, the $5k$ *Unknown* set (29.91) performs on par with the $10k$ *Combine* set (29.57). This pattern validates that KA2L can effectively halve the annotation and computational budget with negligible performance trade-offs by focusing on the most valuable "unknown" data.

Table 6: **Active learning performance on the NQ_Open dataset.** "None" indicates the base model. "5k Known" and "5k Unknown" denote fine-tuning with 5,000 samples from "Known" and KA2L-selected "Unknown" regions, respectively. "10k Combine" is a 5k "Known" + 5k "Unknown" mix, and "10k Unknown" uses 10k KA2L-selected "Unknown" samples. "10k Augmented" involves augmenting "5k Unknown" to 10,000 samples via hidden state decoding, followed by distillation using GPT-4o.

| Model | SFT Dataset | BLEU | ROUGE-L | METEOR | BS(%) | SFT Dataset | BLEU | ROUGE-L | METEOR | BS(%) |
|---|---|---|---|---|---|---|---|---|---|---|
| DeepSeek-R1-Distill-Qwen-7B | None | 0.02 | 0.46 | 1.26 | 76.39 | 10k Combine | 6.10 | 13.17 | 9.68 | 86.45 |
| | 5k Known | 4.69 | 11.15 | 7.87 | 86.19 | 10k Unknown | **7.48** | **16.05** | **11.84** | **86.98** |
| | 5k Unknown | 4.05 | 12.19 | 8.78 | 86.37 | 10k Augmented | 2.85 | 13.56 | 10.04 | 86.49 |
| glm4-9b-chat | None | 0.22 | 3.68 | 8.63 | 79.94 | 10k Combine | 28.78 | 45.94 | 36.50 | 91.33 |
| | 5k Known | 19.55 | 35.78 | 27.67 | 89.62 | 10k Unknown | **39.54** | **57.89** | **46.91** | **93.27** |
| | 5k Unknown | 28.61 | 45.91 | 36.47 | 91.31 | 10k Augmented | 22.49 | 46.00 | 36.62 | 91.34 |
| Llama-2-7b-chat-hf | None | 0.12 | 1.92 | 4.70 | 78.66 | 10k Combine | 21.29 | 39.10 | 31.04 | 90.25 |
| | 5k Known | 12.23 | 25.65 | 19.71 | 88.06 | 10k Unknown | **33.56** | **52.14** | **41.94** | **92.45** |
| | 5k Unknown | 23.47 | 39.25 | 31.15 | 90.23 | 10k Augmented | 17.78 | 38.69 | 30.72 | 90.24 |
| Llama-3.1-8B-Instruct | None | 0.15 | 3.49 | 7.93 | 79.12 | 10k Combine | 24.46 | 45.15 | 35.35 | 91.09 |
| | 5k Known | 16.57 | 32.56 | 24.69 | 89.07 | 10k Unknown | **38.91** | **58.84** | **46.97** | **93.35** |
| | 5k Unknown | 22.32 | 44.94 | 35.04 | 91.08 | 10k Augmented | 21.90 | 44.29 | 34.53 | 91.03 |
| Mistral-7B-Instruct-v0.1 | None | 0.31 | 5.44 | 10.85 | 80.48 | 10k Combine | 21.21 | 35.27 | 27.16 | 89.76 |
| | 5k Known | 13.35 | 26.88 | 20.29 | 88.39 | 10k Unknown | **38.70** | **58.51** | **46.23** | **93.49** |
| | 5k Unknown | 20.44 | 34.59 | 26.69 | 89.56 | 10k Augmented | 14.05 | 35.07 | 27.20 | 89.70 |
| Mistral-7B-Instruct-v0.3 | None | 0.28 | 3.26 | 8.21 | 79.99 | 10k Combine | 28.64 | 46.53 | 36.27 | 91.37 |
| | 5k Known | 16.17 | 33.89 | 25.37 | 89.31 | 10k Unknown | **39.45** | **59.72** | **47.23** | **93.65** |
| | 5k Unknown | 28.30 | 46.28 | 35.99 | 91.44 | 10k Augmented | 24.32 | 46.99 | 36.72 | 91.51 |
| Phi-3.5-mini-instruct | None | 0.15 | 2.11 | 5.42 | 79.18 | 10k Combine | 14.30 | 27.85 | 20.88 | 88.52 |
| | 5k Known | 12.83 | 24.73 | 18.19 | 88.03 | 10k Unknown | **20.01** | **34.98** | **26.71** | **89.66** |
| | 5k Unknown | 14.28 | 27.83 | 20.87 | 88.55 | 10k Augmented | 12.57 | 28.59 | 21.55 | 88.57 |
| Qwen1.5-7B-Chat | None | 0.20 | 2.72 | 6.51 | 79.55 | 10k Combine | 14.97 | 31.62 | 24.45 | 88.89 |
| | 5k Known | 10.12 | 22.56 | 16.98 | 87.31 | 10k Unknown | **23.39** | **39.90** | **31.31** | **90.21** |
| | 5k Unknown | 17.05 | 30.86 | 23.66 | 88.78 | 10k Augmented | 12.98 | 33.72 | 26.53 | 89.28 |
| Qwen2.5-7B-Instruct | None | 0.21 | 3.43 | 7.99 | 79.73 | 10k Combine | 16.23 | 29.57 | 23.12 | 88.34 |
| | 5k Known | 9.81 | 24.26 | 18.68 | 87.51 | 10k Unknown | **17.42** | **34.76** | **27.37** | **89.25** |
| | 5k Unknown | 15.90 | 29.91 | 23.32 | 88.41 | 10k Augmented | 12.79 | 31.67 | 25.10 | 88.70 |

Table 7: **Active learning performance on the TriviaQA dataset.**

| Model | SFT Dataset | BLEU | ROUGE-L | METEOR | BS(%) | SFT Dataset | BLEU | ROUGE-L | METEOR | BS(%) |
|---|---|---|---|---|---|---|---|---|---|---|
| DeepSeek-R1-Distill-Qwen-7B | None | 0.02 | 0.49 | 1.24 | 76.13 | 10k Combine | 9.98 | 23.65 | 16.38 | 87.05 |
| | 5k Known | 7.79 | 18.77 | 12.82 | 86.32 | 10k Unknown | **14.20** | **27.00** | **18.86** | **87.56** |
| | 5k Unknown | 9.41 | 22.22 | 15.39 | 86.85 | 10k Augmented | 6.68 | 24.46 | 17.11 | 87.16 |
| glm4-9b-chat | None | 0.15 | 3.62 | 8.16 | 79.13 | 10k Combine | **21.51** | 41.26 | 30.71 | 88.18 |
| | 5k Known | 14.68 | 31.65 | 23.03 | 86.34 | 10k Unknown | 11.68 | **55.80** | **42.43** | **91.00** |
| | 5k Unknown | 19.62 | 40.28 | 29.85 | 87.97 | 10k Augmented | 15.21 | 40.47 | 30.07 | 88.09 |
| Llama-2-7b-chat-hf | None | 0.09 | 2.30 | 5.24 | 78.23 | 10k Combine | 26.36 | 49.16 | 36.84 | 89.37 |
| | 5k Known | 19.67 | 41.86 | 31.33 | 88.06 | 10k Unknown | **33.25** | **58.91** | **44.77** | **91.58** |
| | 5k Unknown | 24.57 | 48.90 | 36.69 | 89.33 | 10k Augmented | 21.97 | 47.57 | 35.75 | 89.14 |
| Llama-3.1-8B-Instruct | None | 0.12 | 5.67 | 11.11 | 79.26 | 10k Combine | 23.59 | 49.78 | 36.37 | 89.65 |
| | 5k Known | 14.81 | 43.14 | 31.54 | 88.50 | 10k Unknown | **30.84** | **63.38** | **47.05** | **92.39** |
| | 5k Unknown | 18.96 | 49.23 | 35.95 | 89.61 | 10k Augmented | 22.05 | 49.56 | 36.23 | 90.12 |
| Mistral-7B-Instruct-v0.1 | None | 0.32 | 15.73 | 17.64 | 81.85 | 10k Combine | 18.41 | 46.51 | 33.61 | 89.08 |
| | 5k Known | 13.20 | 39.35 | 28.41 | 87.88 | 10k Unknown | **32.77** | **63.64** | **47.43** | **92.42** |
| | 5k Unknown | 15.93 | 45.52 | 32.98 | 88.92 | 10k Augmented | 15.67 | 45.86 | 33.21 | 89.01 |
| Mistral-7B-Instruct-v0.3 | None | 0.16 | 3.14 | 7.11 | 79.07 | 10k Combine | 22.75 | 47.65 | 34.81 | 89.12 |
| | 5k Known | 16.87 | 40.87 | 29.36 | 87.94 | 10k Unknown | **36.84** | **62.41** | **46.66** | **92.21** |
| | 5k Unknown | 21.21 | 47.21 | 34.42 | 89.02 | 10k Augmented | 16.63 | 47.13 | 34.27 | 88.99 |
| Phi-3.5-mini-instruct | None | 0.11 | 2.41 | 5.85 | 78.86 | 10k Combine | 22.93 | 43.39 | 31.60 | 89.01 |
| | 5k Known | 18.13 | 39.10 | 28.31 | 88.29 | 10k Unknown | **25.72** | **47.15** | **34.19** | **89.73** |
| | 5k Unknown | 23.26 | 42.94 | 31.24 | 88.98 | 10k Augmented | 19.26 | 42.98 | 31.27 | 88.99 |
| Qwen1.5-7B-Chat | None | 0.14 | 3.84 | 8.18 | 79.28 | 10k Combine | 20.78 | 38.85 | 28.16 | 88.39 |
| | 5k Known | 20.04 | 39.05 | 28.53 | 88.40 | 10k Unknown | **25.91** | **48.38** | **35.80** | **90.03** |
| | 5k Unknown | 19.89 | 39.02 | 28.54 | 88.40 | 10k Augmented | 13.85 | 38.06 | 27.56 | 88.24 |
| Qwen2.5-7B-Instruct | None | 0.15 | 4.07 | 8.79 | 78.92 | 10k Combine | 11.37 | 32.52 | 24.36 | 85.65 |
| | 5k Known | 11.72 | 32.24 | 24.18 | 85.63 | 10k Unknown | **14.49** | **35.88** | **27.07** | **86.49** |
| | 5k Unknown | 11.72 | 32.30 | 24.20 | 85.63 | 10k Augmented | 5.82 | 31.99 | 24.24 | 85.80 |

Table 8: **Performance comparison of active learning methods on the TriviaQA dataset.** Our method, KA2L (**5k Unknown**), significantly outperforms all traditional methods. Results are reported as mean ± std over 4 runs. The best result among 5k-sample methods is in **bold**.

| Method | BLEU | ROUGE-L | METEOR | BertScore(%) |
|---|---|---|---|---|
| *Traditional Methods (5k samples selected from 10k pool)* | | | | |
| Random | $12.33 \pm 0.14$ | $45.25 \pm 0.01$ | $32.96 \pm 0.02$ | $88.95 \pm 0.01$ |
| Entropy | $\mathbf{22.34 \pm 0.08}$ | $46.25 \pm 0.09$ | $33.91 \pm 0.06$ | $89.10 \pm 0.01$ |
| Coreset | $20.37 \pm 1.80$ | $45.64 \pm 0.07$ | $33.31 \pm 0.04$ | $89.02 \pm 0.00$ |
| BADGE (adapted) | $20.58 \pm 0.22$ | $45.50 \pm 0.06$ | $33.15 \pm 0.03$ | $88.98 \pm 0.01$ |
| *Our Method (5k samples)* | | | | |
| KA2L 5k Known | $15.88 \pm 1.65$ | $43.19 \pm 0.05$ | $31.55 \pm 0.03$ | $88.51 \pm 0.01$ |
| **KA2L 5k Unknown** | $18.92 \pm 0.41$ | $\mathbf{49.21 \pm 0.03}$ | $\mathbf{35.93 \pm 0.02}$ | $\mathbf{89.61 \pm 0.00}$ |
| *Upper Bound* | | | | |
| Full Dataset (10k) | $23.50 \pm 0.35$ | $49.76 \pm 0.06$ | $36.33 \pm 0.05$ | $89.65 \pm 0.01$ |

Table 9: **Performance comparison of active learning methods on the MedMCQA dataset.**

| Method | BLEU | ROUGE-L | METEOR | BertScore(%) |
|---|---|---|---|---|
| *Traditional Methods (5k samples selected from 10k pool)* | | | | |
| Random | $7.19 \pm 0.22$ | $26.67 \pm 0.05$ | $18.55 \pm 0.05$ | $86.44 \pm 0.02$ |
| Entropy | $6.85 \pm 0.09$ | $26.66 \pm 0.07$ | $18.46 \pm 0.05$ | $86.40 \pm 0.01$ |
| Coreset | $8.11 \pm 0.11$ | $28.48 \pm 0.08$ | $19.86 \pm 0.03$ | $86.83 \pm 0.01$ |
| BADGE (adapted) | $6.68 \pm 0.12$ | $26.76 \pm 0.06$ | $18.64 \pm 0.07$ | $86.41 \pm 0.01$ |
| *Our Method (5k samples)* | | | | |
| KA2L 5k Known | $5.67 \pm 0.06$ | $23.35 \pm 0.05$ | $16.23 \pm 0.03$ | $85.73 \pm 0.01$ |
| **KA2L 5k Unknown** | $\mathbf{8.47 \pm 0.14}$ | $\mathbf{29.96 \pm 0.02}$ | $\mathbf{20.84 \pm 0.01}$ | $\mathbf{87.19 \pm 0.01}$ |
| *Upper Bound* | | | | |
| Full Dataset (10k) | $8.17 \pm 0.04$ | $30.14 \pm 0.03$ | $21.08 \pm 0.02$ | $87.13 \pm 0.01$ |

**Effectiveness of Targeted Selection (RQ2).** The superiority of KA2L's selection strategy is unequivocally confirmed when comparing datasets of the same size. The $10k$ *Unknown* setting consistently and significantly outperforms the $10k$ *Combine* baseline across nearly all models and metrics. On NQ_Open, for example, `Qwen2.5-7B-Instruct` achieves a ROUGE-L of 34.76 with $10k$ *Unknown* data, a substantial 5.19-point improvement over the 29.57 from the $10k$ *Combine* set. On TriviaQA, `DeepSeek-R1-Distill-Qwen-7B` shows a similar leap, from 23.65 ($10k$ *Combine*) to 27.00 ($10k$ *Unknown*). These results provide strong evidence that, for a fixed budget, intelligently selecting data that addresses a model's knowledge gaps is far more effective than an unfiltered, naive training approach.

**Utility of Data Augmentation (RQ3).** The $10k$ *Augmented* strategy shows practical utility, though its effectiveness varies compared to the main experiment on MedMCQA. As expected, augmented data generally improves performance over the initial $5k$ *Unknown* set but does not reach the ceiling set by the $10k$ *Unknown* set. For example, with `DeepSeek-R1-Distill-Qwen-7B` on NQ_Open, the ROUGE-L score improves from 12.19 ($5k$ *Unknown*) to 13.56 ($10k$ *Augmented*), but falls short of the 16.05 achieved by $10k$ *Unknown*. This outcome suggests that for broad, open-domain datasets, the diversity of original data is paramount. While our hidden-state decoding method provides a clear benefit, the augmented samples may not introduce the same breadth of novel knowledge as genuinely new "unknown" samples, in contrast to the more focused knowledge space of a specialized domain like MedMCQA. This highlights a potential area for future research in adapting augmentation techniques for open-domain contexts.

### F.2 FULL RESULTS FOR TRADITIONAL ACTIVE LEARNING METHODS COMPARISON

This section provides the complete results of our comparison against traditional active learning methods on the TriviaQA and MedMCQA datasets, supplementing the primary analysis on NQ_Open presented in the main paper. The experiments were conducted on the `LLaMA-3.1-8B-Instruct` model, with each method selecting 5,000 samples from a $10k$ data pool. The results are detailed in Table 8 and Table 9.

Table 10: **Performance comparison of knowledge distribution probes (AUROC) on the NQ_Open dataset.**

| Models | Ours | SE Probe | Accuracy Probe | Log-Likelihood | Regular Entropy | P(True) | Semantic Entropy |
|---|---|---|---|---|---|---|---|
| DeepSeek-R1-Distill-Qwen-7B | **0.88** | 0.84 | 0.70 | 0.63 | 0.72 | 0.81 | 0.80 |
| glm4-9b-chat | **0.83** | **0.83** | 0.70 | 0.56 | 0.72 | 0.79 | 0.76 |
| Llama-2-7b-chat-hf | **0.82** | 0.76 | 0.68 | 0.55 | 0.70 | 0.74 | 0.73 |
| Llama-3.1-8B-Instruct | **0.83** | 0.81 | 0.72 | 0.58 | 0.71 | 0.77 | 0.77 |
| Mistral-7B-Instruct-v0.1 | **0.88** | 0.84 | 0.75 | 0.64 | 0.70 | 0.77 | 0.78 |
| Mistral-7B-Instruct-v0.3 | **0.85** | 0.81 | 0.70 | 0.65 | 0.72 | 0.77 | 0.77 |
| Phi-3.5-mini-instruct | **0.91** | 0.87 | 0.76 | 0.76 | 0.80 | 0.83 | 0.84 |
| Qwen1.5-7B-Chat | **0.84** | 0.78 | 0.71 | 0.46 | 0.66 | 0.79 | 0.71 |
| Qwen2.5-7B-Instruct | **0.86** | 0.83 | 0.72 | 0.55 | 0.76 | 0.80 | 0.80 |

Table 11: **Performance comparison of knowledge distribution probes (AUROC) on the MedMCQA dataset.**

| Models | Ours | SE Probe | Accuracy Probe | Log-Likelihood | Regular Entropy | P(True) | Semantic Entropy |
|---|---|---|---|---|---|---|---|
| DeepSeek-R1-Distill-Qwen-7B | **0.81** | 0.77 | 0.73 | 0.62 | 0.77 | 0.80 | 0.79 |
| glm4-9b-chat | **0.81** | **0.81** | 0.71 | 0.53 | 0.65 | 0.73 | 0.70 |
| Llama-2-7b-chat-hf | **0.81** | 0.76 | 0.70 | 0.54 | 0.67 | 0.67 | 0.69 |
| Llama-3.1-8B-Instruct | **0.84** | 0.70 | 0.78 | 0.57 | 0.69 | 0.79 | 0.74 |
| Mistral-7B-Instruct-v0.1 | **0.85** | 0.81 | 0.73 | 0.59 | 0.68 | 0.70 | 0.72 |
| Mistral-7B-Instruct-v0.3 | **0.85** | 0.80 | 0.73 | 0.61 | 0.68 | 0.75 | 0.72 |
| Phi-3.5-mini-instruct | **0.80** | 0.78 | 0.70 | 0.64 | 0.70 | 0.73 | 0.73 |
| Qwen1.5-7B-Chat | **0.79** | 0.72 | 0.68 | 0.48 | 0.65 | 0.76 | 0.66 |
| Qwen2.5-7B-Instruct | 0.76 | 0.70 | 0.71 | 0.55 | 0.71 | **0.78** | 0.75 |

The results on both TriviaQA and MedMCQA robustly corroborate the findings presented in the main text. On TriviaQA (Table 8), fine-tuning on the $5k$ *Unknown* set selected by KA2L achieves a ROUGE-L of $49.21$, decisively outperforming the strongest traditional method (Entropy, $46.25$) by nearly 3 points. Similarly, on the specialized MedMCQA dataset (Table 9), KA2L's selection yields a ROUGE-L of $29.96$, a significant 1.5-point gain over the best-performing method, Coreset ($28.48$).

Crucially, this superior performance is achieved with remarkable cost-efficiency. On both datasets, the $5k$ *Unknown* set approaches the performance of the *Full Dataset* ($10k$), which uses double the data. For MedMCQA, the performance is nearly identical ($29.96$ vs. $30.14$ ROUGE-L). Furthermore, the consistently poor performance of the $5k$ *Known* set, which often scores below random sampling, strongly validates KA2L's ability to effectively partition data into mastered and unmastered knowledge, thereby filtering out redundant samples. These comprehensive results across diverse domains confirm that KA2L's knowledge-centric selection strategy is a more effective and efficient approach for LLM fine-tuning compared to traditional AL methods.

## F.3 KNOWLEDGE DISTRIBUTION PROBE PERFORMANCE EVALUATION

The experimental results presented in Table 10 for the NQ_Open dataset highlight the efficacy of our proposed knowledge distribution probe. Our method **consistently achieves the highest** Area Under the ROC Curve (AUROC) (Ling et al., 2003) across nearly all evaluated models, signifying superior accuracy in distinguishing between known and unknown information. For example, our probe attains an AUROC of $0.91$ for the `Phi-3.5-mini-instruct` model and $0.88$ for both `DeepSeek-R1-Distill-Qwen-7B` and `Mistral-7B-Instruct-v0.1`. This represents a consistent improvement over the SE Probe; for instance, with `Llama-2-7b-chat-hf`, our method scores $0.82$ compared to SE Probe's $0.76$, and for `Qwen1.5-7B-Chat`, the scores are $0.84$ versus $0.78$. Furthermore, our approach demonstrates substantial gains over traditional uncertainty metrics such as Log-Likelihood and Regular Entropy. Compared to P(True)

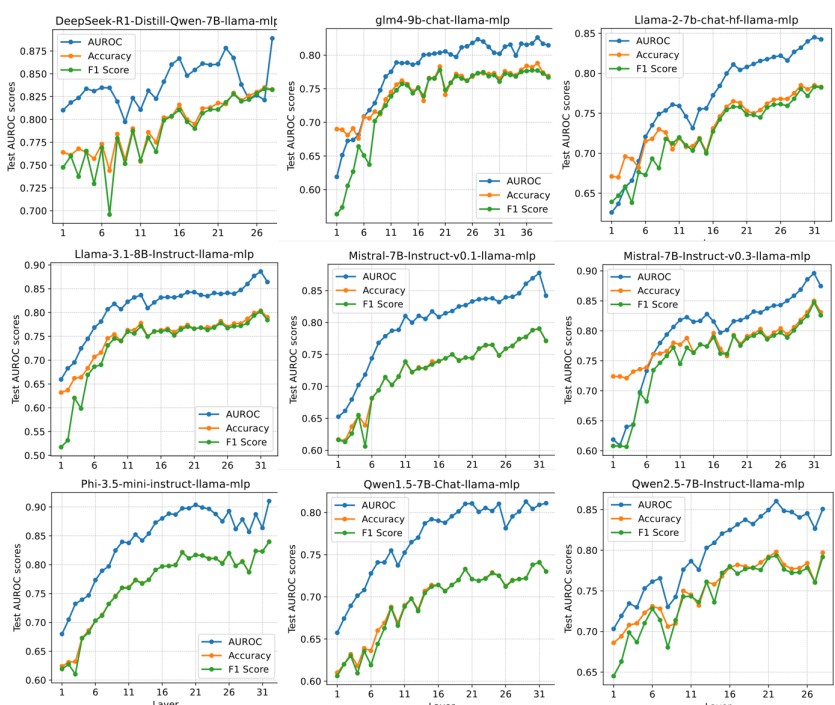

Figure 2: **Classifier AUROC score across different hidden layers** (embedding layer excluded) on the Trivia_QA dataset.

and Semantic Entropy, our probe generally exhibits higher AUROC values; for example, with `Llama-3.1-8B-Instruct`, our method achieves 0.83, while P(True) and Semantic Entropy score 0.77.

Similar performance advantages for our proposed probe are evident on the MedMCQA dataset, as detailed in Table 11. Our method again **achieves the top AUROC scores** for the majority of models, such as 0.85 for `Mistral-7B-Instruct-v0.1` and `Mistral-7B-Instruct-v0.3`, and 0.84 for `Llama-3.1-8B-Instruct`. It is worth noting that AUROC values across all methods are often slightly lower on MedMCQA compared to NQ_Open, potentially reflecting the distinct characteristics or increased complexity of this specialized medical domain for knowledge probing. The improvement of our method over SE Probe remains consistent, with a particularly notable margin for `Llama-3.1-8B-Instruct` (0.84 for ours vs. 0.70 for SE Probe) and `Qwen1.5-7B-Chat` (0.79 vs. 0.72). As with the NQ_Open dataset, our probe significantly outperforms Log-Likelihood and Regular Entropy across all models. When compared against P(True) and Semantic Entropy, our method generally maintains a performance lead. For example, on `DeepSeek-R1-Distill-Qwen-7B`, our probe scores 0.81, whereas P(True) and Semantic Entropy achieve 0.80 and 0.79, respectively. A single exception is noted for `Qwen2.5-7B-Instruct`, where P(True) (0.78) marginally surpasses our method (0.76). Overall, these results underscore the robust and superior classification capability of our improved MLP-based probe across different datasets and a diverse set of language models.

## F.4 KNOWLEDGE DISTRIBUTION PROBE LAYER-WISE ANALYSIS

To identify the most effective source of representations for our knowledge distribution probe, we conducted a comprehensive analysis of classifier performance across the hidden layers of various LLMs. The optimal hidden layer identified for each LLM served as a crucial basis for selecting the input for the probe and decoder in our main experiments. This analysis also reveals insights into how uncertainty-related information is encoded throughout these models.

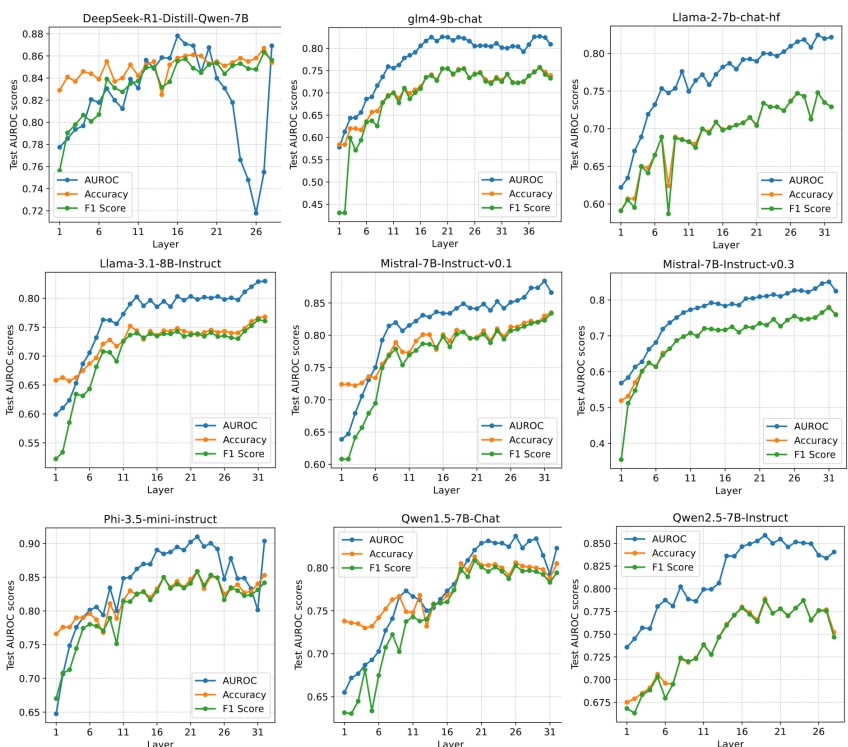

Figure 3: **Classifier AUROC score across different hidden layers** (embedding layer excluded) on the NQ_Open dataset.

We evaluated the performance of classifiers trained on different hidden layer representations (excluding the embedding layer) for nine distinct models across three datasets. The results are presented in Figure 2 (TriviaQA), Figure 3 (NQ_Open), and Figure 4 (MedMCQA). A consistent, overarching trend observed across all datasets and most models is that classifier performance, measured by AUROC, generally improves with increasing layer depth. This suggests that in short-answer QA tasks like TriviaQA, deeper hidden layers may contain more features pertinent to output uncertainty.

Beyond this general trend, we identified distinct, architecture-specific patterns that are remarkably consistent across datasets. A notable example is the behavior of DeepSeek-R1-Distill-Qwen-7B and Qwen2.5-7B-Instruct, which both display a complex pattern: after an initial rise, their performance shows a discernible dip in the final few layers. This shared characteristic, observed on all three datasets, suggests a common architectural trait. In contrast, Phi-3.5-mini-instruct demonstrates rapid performance gains in earlier layers, with its AUROC tending to plateau in the deeper layers. These consistent, model-specific signatures reinforce the hypothesis that the encoding of uncertainty is closely tied to model architecture.

To further disentangle the influence of model architecture from that of the dataset, we analyzed the performance of a single model, Llama-3.1-8B-Instruct, across all three datasets (Figure 5). The results show that the layer-wise AUROC curve maintains a very similar shape regardless of the dataset. This finding suggests that the variation in detection accuracy across layers is less influenced by the specific knowledge domain of the dataset and is more fundamentally correlated with the intrinsic properties of the model's architecture. The precise mechanisms governing these layer-wise patterns warrant further investigation.

## F.5    IMPACT OF UNKNOWN DATA VOLUME ON FINE-TUNING PERFORMANCE

To further validate our active learning strategy, we investigated the impact of data volume on model performance. We fine-tuned the Llama-3.1-8B-Instruct model on the MedMCQA dataset using incrementally larger subsets of "unknown" data identified by KA2L, with results shown in Figure 6.

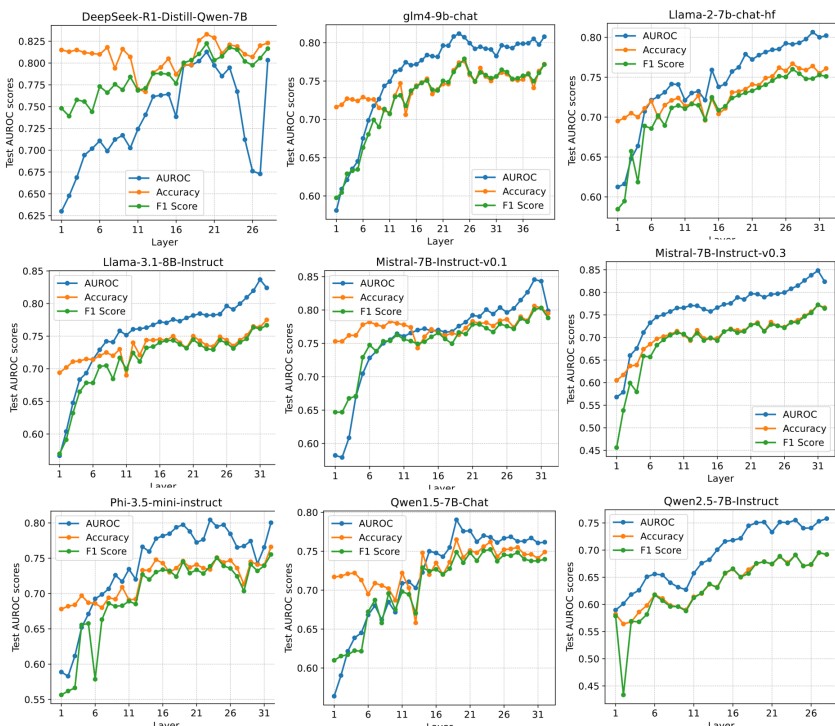

Figure 4: **Classifier AUROC score across different hidden layers** (embedding layer excluded) on the MedMCQA dataset.

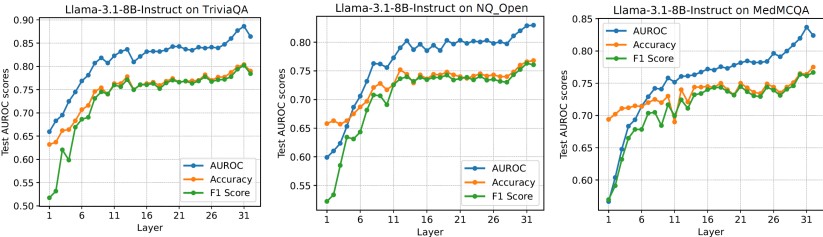

Figure 5: Llama3.1 classifier AUROC score across hidden layers on TriviaQA, NQ_Open, and MedMCQA datasets

The analysis reveals a strong, positive correlation between the quantity of "unknown" data and the model's performance across all metrics. This trend confirms that the data selected by our framework is consistently informative, and performance scales effectively with the amount of high-value knowledge selected. A key observation is the high marginal utility of the initial data samples. For instance, on the ROUGE-L metric, the first $5,000$ samples contribute a substantial portion (over $80\%$) of the total performance gain achieved with $10,000$ samples. This finding informed our choice of the $5k$ sample size in our main experiments as an effective point to demonstrate a favorable trade-off between performance and cost-efficiency.

Unlike traditional active learning on closed-set classification tasks, where performance often saturates quickly, our results exhibit a more sustained, near-linear growth. This characteristic is attributable to the open-ended nature of the knowledge-intensive QA task and the vast capacity of large language models. The continuous performance improvement suggests that our KA2L framework is highly effective at persistently identifying non-redundant, novel knowledge points from a large pool, a desirable property for any active learning system designed for continual knowledge acquisition. The consistent gains across different data volumes underscore the robustness and efficacy of our semantic entropy-based selection method.

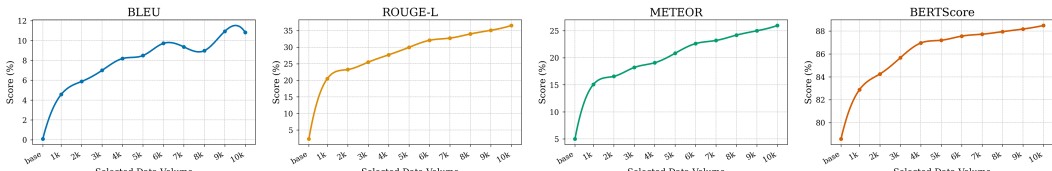

Figure 6: Fine-tuning performance of Llama-3.1-8B-Instruct on the MedMCQA dataset with varying amounts of KA2L-selected "unknown" data. The x-axis represents the number of training samples, while the "base" point indicates the model's performance without fine-tuning.

Table 12: AUROC scores of the knowledge distribution probe under perturbed binarization thresholds for `LLaMA-3.1-8B-Instruct` (Layer 31). The optimal threshold $\gamma^*$ is determined by minimizing MSE. The highest score in each row is highlighted in bold.

| Dataset | $\gamma^* - 0.20$ | $\gamma^* - 0.10$ | $\gamma^* - 0.05$ | $\gamma^*$ | $\gamma^* + 0.05$ | $\gamma^* + 0.10$ | $\gamma^* + 0.20$ |
|---|---|---|---|---|---|---|---|
| TriviaQA | 0.8543 | 0.8602 | 0.8623 | **0.8853** | 0.8847 | 0.8784 | 0.8721 |
| NQ_Open | 0.8232 | 0.8290 | 0.8277 | **0.8307** | 0.8265 | 0.8248 | 0.8247 |
| MedMCQA | 0.8281 | 0.8267 | 0.8323 | **0.8345** | 0.8285 | 0.8238 | 0.8333 |

### F.6 ROBUSTNESS ANALYSIS OF THE DYNAMIC THRESHOLD

To assess the robustness of our dynamic thresholding method for binarizing Semantic Entropy, we conducted a perturbation analysis. The method's objective is to adaptively find an optimal threshold, $\gamma^*$, by minimizing the MSE between continuous SE values and their binarized counterparts, as detailed in Section 3 (Equation 4).

For this analysis, we focused on the `LLaMA-3.1-8B-Instruct` model, using hidden states from its optimal 31st layer. We performed experiments on the TriviaQA, NQ_Open, and MedMCQA datasets. The optimal thresholds $\gamma^*$ identified by our method were 0.954, 1.326, and 1.419 for TriviaQA, NQ_Open, and MedMCQA, respectively. We then perturbed this optimal threshold by increments of $\pm 0.05$, $\pm 0.10$, and $\pm 0.20$, re-trained the knowledge distribution probe with the newly binarized labels, and evaluated its performance (AUROC) on the test set.

The results, presented in Table 12, show that the classifier consistently achieves the highest AUROC score precisely at the optimal threshold $\gamma^*$ across all three datasets. This empirically validates that our MSE-based method is effective at identifying an optimal cut-off point. Furthermore, the performance degrades gracefully as the threshold deviates from the optimum. Even with a significant perturbation, the performance does not collapse, indicating that our framework is not overly sensitive to the exact threshold value and demonstrates practical robustness.

### F.7 QUALITATIVE ANALYSIS

To supplement our quantitative findings and provide a more intuitive validation of our core mechanism, we present a qualitative case study. The central premise of the KA2L framework is that an LLM's mastery of a knowledge point can be effectively proxied by the semantic consistency of its generated answers. This section aims to visually substantiate this premise by examining model outputs across different Semantic Entropy (SE) ranges.

**Low Entropy as an Indicator of "Known" Knowledge.** As demonstrated in Table 13, questions with low SE ($\approx 0$) consistently elicit answers that are not only correct but also exhibit high semantic stability. For instance, when asked about the number of keys on a standard keyboard, the model reliably generates "104" across all ten samples. Minor lexical variations, such as between "John Lennon" and "john lennon", do not alter the core semantic meaning, resulting in a tight cluster of answers and consequently, a low SE score. This pattern vividly illustrates that low semantic entropy is a strong signal of mastered, stable knowledge. It underscores the validity of our operational definition and justifies KA2L's strategy of identifying these samples as "known" to avoid redundant and inefficient fine-tuning.

**High Entropy as an Indicator of "Unknown" Knowledge.** In stark contrast, Table 15 showcases the behavior for high-entropy ($\approx 2.3$) questions. When faced with a query it has not mastered, such as "who holds the record for most knockouts in boxing", the model's outputs become chaotic and semantically divergent. The generated answers span a wide range of incorrect names ("Rocky Marciano," "Muhammad Ali," "George Foreman") and even nonsensical phrases, indicating a clear lack of confident knowledge. This high degree of semantic inconsistency directly translates to a high SE score. This observation provides strong qualitative evidence that high semantic entropy is a reliable indicator of "unknown" knowledge, thereby validating KA2L's core principle of actively selecting these high-value samples for targeted fine-tuning.

**The Knowledge Boundary.** The intermediate cases, shown in Table 14, are equally revealing. For questions with medium SE scores, the model's outputs are a mixture of correct, partially correct, and incorrect answers. For example, when asked about the origin of Häagen-Dazs, the model generates the correct answer "United States" but also plausible yet incorrect alternatives like "Poland" and "Scandinavia." This instability reflects a state of epistemic uncertainty—the model is at the "knowledge boundary" where it may have encountered the information but has not fully assimilated it. These cases highlight that SE effectively captures the continuous spectrum of knowledge mastery, from confidently known to completely unknown, making it a robust signal for guiding the active learning process.

Table 13: Case Study: Low Entropy Samples (SE $\approx 0$)

| # | Question | Ground Truth | Generated Answers (10 Samples) |
|---|---|---|---|
| 1 | what is the size of the angles of an equilateral triangle | 60° | 60
60
60
60
60
60 degrees
60
60
60
60 degrees |
| 2 | who played willy wonka in the original movie | Gene Wilder | Gene Wilder
Gene Wilder
gene wilder
Gene Wilder
Gene Wilder
Gene Wilder
Gene Wilder
Gene Wilder
Gene Wilder
Gene wilder |
| 3 | how many keys are on microsoft windows standard keyboard | 104 | 104
104
104
104
104
104
104
104
104
104 |

Table 13: (Continued) Case Study: Low Entropy Samples (SE ≈ 0)

| # | Question | Ground Truth | Generated Answers (10 Samples) |
|---|----------|--------------|-------------------------------|
| 4 | who wrote the lyrics to all you need is love | John Lennon | `John Lennon`
`John Lennon`
`john lennon`
`John Lennon`
`John Lennon`
`John Lennon`
`john lennon`
`Lennon`
`John Lennon`
`John Lennon` |
| 5 | what was the robot maids name on the jetsons | Rosie | `Rosie`
`Rosie`
`Rosie`
`Rosie`
`Rosie`
`Rosie`
`Rosie`
`Rosie`
`Rosie`
`Rosie` |
| 6 | when did wrath of the lich king release | November 13, 2008 | `November 13, 2008`
`November 13, 2008`
`November 13, 2008`
`November 13, 2008`
`November 13, 2008`
`November 13, 2008`
`November 13, 2008`
`November 13, 2008`
`November 13, 2008`
`November 13, 2008` |
| 7 | what does the zip in zip code stand for | Zone Improvement Plan | `Zone Improvement Plan`
`Zone Improvement Plan`
`Zone Improvement Plan`
`Zone Improvement Plan`
`zone improvement plan`
`zone improvement plan`
`Zoning Improvement Plan`
`Zone Improvement Plan`
`Zone Improvement Plan`
`zone improvement plan` |
| 8 | movie with will smith and tommy lee jones | Men in Black | `men in black`
`Men in Black`
`Men in Black`
`Men in Black`
`men in black`
`Men in Black`
`Men in Black`
`Men in Black`
`men in black`
`Men in Black` |

Table 13: (Continued) Case Study: Low Entropy Samples (SE ≈ 0)

| # | Question | Ground Truth | Generated Answers (10 Samples) |
|---|----------|--------------|-------------------------------|
| 9 | how many original harry potter books are there | 7 | `Seven`
`7`
`7`
`7`
`Seven`
`7`
`7`
`seven`
`7`
`7` |
| 10 | what is the city in italy that is under water | Venice | `Venice`
`venice`
`venice`
`Venice`
`Venice`
`Venice`
`venice`
`Venice`
`Venice`
`venice` |

Table 14: Case Study: Medium Entropy Samples (SE ≈ 1.3 - 1.7)

| # | Question | Ground Truth | Generated Answers (10 Samples) |
|---|----------|--------------|-------------------------------|
| 1 | who wins americas next top model season 18 | Sophie Sumner | `Sophie Sumner`
`Sophie Sumner`
`Sophie Sumner`
`Sophie Sumner`
`Erica Ash`
`Laura James`
`Sophia L?Hommerieux`
`Laura James`
`Lisa DAmato`
`Sophie Sumner` |
| 2 | star left at the core of a planetary nebula | white dwarf | `exhausted carbon...`
`Carbon core`
`Other`
`white dwarf`
`dead star`
`a white dwarf`
`White Dwarf`
`dead star`
`white dwarf`
`white dwarf` |
| 3 | who has the most wins ever in nascar | Richard Petty | `Richard Petty (200)...`
`Richard Petty`
`Richard Petty`
`dale earnhardt with (76)"`
`jeff gordon`
`jeff gordon`
`Richard Petty`
`stylsmilian roger penske`
`jeff gordon`
`jeff gordon` |

Table 14: (Continued) Case Study: Medium Entropy Samples (SE ≈ 1.3 - 1.7)

| # | Question | Ground Truth | Generated Answers (10 Samples) |
|---|----------|--------------|-------------------------------|
| 4 | who said what's done is done in macbeth | Lady Macbeth | Macduff
Macbeth
Macduff
Donalbain
lady macbeth
lady macbeth
The dagger on...
Macduff
Lady Macbeth
Lady Macbeth |
| 5 | where does the pulmonary artery take blood to | the lungs | right ventricle
the pulmonary capillaries
the left atrium
The left atrium
...oxygen-rich blood...
the left atrium
lungs
the lungs
The lungs
Lung |
| 6 | the haagen-dazs brand of ice cream originated in which country | United States | United States
Poland
United States
Scandinavia
Poland
Poland
scandinavian
Belgium
Poland
Jamaica |
| 7 | who has the most superbowl wins in nfl history | Pittsburgh Steelers | Tom Brady
pittsburgh steelers.
Six -type is not...
Pittsburgh Steelers
pittsburgh steelers have 6
Pittsburgh Steelers
...Green Bay Packers...
Pittsburgh Steelers with six
Tom Brady
Pittsburgh Steelers |
| 8 | when did the toronto blue jays win the world series | 1993 | There is no correct answer
1992
They have won none
1992
None, they have not...
there has been no...
yes
1992 and 1993
None, they have not
1992 |

Table 14: (Continued) Case Study: Medium Entropy Samples (SE ≈ 1.3 - 1.7)

| # | Question | Ground Truth | Generated Answers (10 Samples) |
|---|----------|--------------|-------------------------------|
| 9 | what are the main components of earth's atmosphere | nitrogen | `Nitrogen, Oxygen, and Argon`
`nitrogen, oxygen, and...`
`Nitrogen Oxygen Carbon...`
`Nitrogen, Oxygen, Argon...`
`nitrogen oxygen argon...`
`Nitrogen, oxygen and argon`
`Nitrogen, Oxygen, Argon`
`Nitrogen, oxygen and argon`
`nitrogen, oxygen & trace...`
`Nitrogen, Oxygen, Argon...` |
| 10 | where do purple martins go in the winter time | South America | `Brazil, Peru, Argentina...`
`Brazil & Argentina`
`Mexico`
`South America`
`South America`
`Brazil`
`South America`
`Brazil and Argentina`
`South america`
`Brazil` |

Table 15: Case Study: High Entropy Samples (SE ≈ 2.3)

| # | Question | Ground Truth | Generated Answers (10 Samples) |
|---|----------|--------------|-------------------------------|
| 1 | when did tonight's going to be a good night come out | June 23, 2009 | `June 14, 2013`
`September 2, 2016`
`January 1, 2010`
`October 1, 2012`
`2009`
`March 22, 2017`
`9 january 2016`
`March 27, 2013`
`2018`
`2017` |
| 2 | where did they film the cowboys with john wayne | New Mexico | `california`
`boston texas`
`Sedona Arizona`
`Monument Valley, Arizona`
`Arizona & California`
`texas`
`Arizona and Utah`
`Bakersfield, California...`
`Arizona`
`Monument Valley` |
| 3 | who is the actual president of the supreme court of justice | John G. Roberts | `Joy Bivoni`
`kennedy library`
`Justice enrique...`
`María Adelina Díaz...`
`Silvia Bercioux`
`Juan Cueto`
`Enrique Iglesias`
`Problem unknown`
`Manuel Marín Perales`
`an acting president is...` |

Table 15: (Continued) Case Study: High Entropy Samples (SE ≈ 2.3)

| # | Question | Ground Truth | Generated Answers (10 Samples) |
|---|----------|--------------|-------------------------------|
| 4 | when was the dung beetle introduced to australia | 1965 | 1895
November 2002
...in the 18th century
1887
the early 1970s
1933
1930
May 1, 2020
Not before 1769...
1959 |
| 5 | who holds the record for most knockouts in boxing | Archie Moore | Rocky Marciano &...
Chris Byrd doesn't
Muhammad Ali
Jabbar Ali
Tyson
Joe Louis
Joe Louis *482*...
Partinello Wolf
George Foreman
Yvon Neptune |
| 6 | who was killed in i know what you did last summer | a pedestrian | Brendan?Fraser
ray bronson
Benny
angelo brandt
David Egan Schwimmer
Julie James
Cue
Davey
Billy and Ray Bronson
Joe White |
| 7 | what is the average salary of a working american | $59,039 | Per computronics 71,301
$47,940
$58,715
$54,000
$55,292
$48,232.42
$74841 USD
$50,708 in May 2021
$64,968
$46,641 |
| 8 | who wrote another somebody done somebody wrong song | Larry Butler and Chips Moman | Hee Haw Lee Hayes
song writers
Bo Diddley
Ronn Gibbs
Gordon Lightfoot.
Baileys
Carl Nugent
Suspicious People...
wedding planner
George Jones |

Table 15: (Continued) Case Study: High Entropy Samples (SE ≈ 2.3)

| # | Question | Ground Truth | Generated Answers (10 Samples) |
|---|----------|--------------|-------------------------------|
| 9 | who played molly ringwalds father in pretty in pink | Harry Dean Stanton | Annie Potts was...
Moody other son...
Harry Dean Stanton
alfie wise
timothy busfield
Rob Lowe
...Duckie, played by...
Annie Potts and...
moody
Harry Dean Stanton... |
| 10 | whats the clown's name in house of 1000 corpses | Captain Spaulding | Otis
Otis B Draught
Blanky
Bill and Colleen
Traumatone
dr listen
Otto and Juno
Ursula, also played...
Stanton Duckworth...
Mother Superior |

