# OpenReview forum: "KA2L: A Knowledge-Aware Active Learning Framework for LLMs"
_ICLR.cc/2026/Conference — ICLR 2026 Conference Withdrawn Submission_

### Official Review · Reviewer_SJ6L · 2025-10-25

**Soundness:** 2
**Presentation:** 3
**Contribution:** 2
**Rating:** 4
**Confidence:** 3

**Summary:**

This paper proposes an active learning method named KA2L, which leverages Semantic Entropy (SE) on hidden representations and the output of LLM responses to determine whether the model possesses knowledge about a given input query. It then fine-tunes the LLM using (query, answer) pairs that are deemed unknown to the model.

**Strengths:**

The paper is clearly presented, and the proposed method is well-explained. Additional details in the appendix enhance the paper’s credibility.
The underlying motivation is clear and well-justified.

**Weaknesses:**

1. The research contributions are not clearly articulated. Contributions 1 and 2 (lines 079–087) both claim novelty in the proposed framework, particularly in integrating LLM knowledge distribution probing with hallucination detection. However, Contribution 2 appears to be a subset of Contribution 1. It remains unclear which components of the framework are truly novel—for example, the use of SE, or the extraction and concatenation of the last token’s representations across layers.
2. The experiments lack comprehensiveness. There should be a comparison of efficiency and computational cost between KA2L and the baselines. This is especially important given that KA2L only marginally outperforms the baseline Coreset in BLEU (22.29 vs. 22.04) and BERTScore (91.08% vs. 90.06%), as shown in Table 3.
3. KA2L relies on a threshold to determine whether questions are KNOWN or UNKNOWN to the model. This threshold is rigid and costly to estimate.
4. The concept of this threshold—and the binary partitioning of questions into KNOWN and UNKNOWN—is not well justified. A small change in the threshold could easily flip a question from one category to the other.

**Questions:**

1. What is the justification for using the last token’s representations across all layers? Shouldn’t only the representation from the final layer, which is used for decoding and generating answers, be considered?
2. Lines 124–125 state: “Q_{unk} represents the set of questions for which the model's answers are uncertain.” What happens if the LLM provides answers with high certainty that are actually incorrect? Is this scenario considered? Would such questions be classified as KNOWN or UNKNOWN?
3. Where is the definition of SE+i mentioned in line 208?

---

### Official Review · Reviewer_tzEH · 2025-10-31

**Soundness:** 2
**Presentation:** 3
**Contribution:** 2
**Rating:** 4
**Confidence:** 3

**Summary:**

This work introduces a novel method for selecting valuable samples for LLM training based on answer consistency, splitting the unlabeled dataset into known and unknown samples. In addition, a novel augmentation technique was proposed that further enhances model performance with limited resources.

**Strengths:**

- The paper performs robust evaluation in its experiments on three datasets with nine different LLMs.
- Clear Guidance through the experimental section via research questions.
- The paper proposes a simple yet efficient approximation scheme for "Knowledge" of the LLM via a binary MLP classifier and proposes a method for hyperparameter selection.

**Weaknesses:**

- The proposed method is compared to well-known AL methods. However, the setting doesn't really involve an iterative selection scheme but a one-time selection, which does not favor any of the compared AL strategies and leads to heavy overlap in informativeness of selected samples (one-time selection of 5000 samples out of 10000 samples). As a result, the provided empirical evidence in sec. 5.3 is rather meaningless.
- While the adaptation of CoreSet seems appropriate, the adaptation of BADGE seems misleading, adding to the confusion of the first weakness.
- Other, newer, and more powerful DAL strategies have not been considered, especially strategies that can work on hidden representations (e.g., TypiClust).

**Questions:**

- What was the motivation behind framing this work as an active learning method? I would see this more in a filtering-methodology.
- Methods should be compared to known SOTA methods, where BADGE and especially CoreSets have been replaced several years ago. Did you consider other methods?
- Is it possible to perform classic active learning with multiple iterations and updating the representations the strategies work with for a fairer comparison?

---

### Official Review · Reviewer_EJNQ · 2025-10-31

**Soundness:** 3
**Presentation:** 2
**Contribution:** 2
**Rating:** 4
**Confidence:** 3

**Summary:**

This paper proposes KA2L, a knowledge-aware active-learning framework that probes the hidden states of an LLM to partition questions into “known” and “unknown”, then fine-tunes only on the unknown subset plus decoder-generated augmentations. Across nine open-source models and three QA datasets the method cuts annotation/compute costs by ~50 % while matching or exceeding full-data performance, and consistently outperforms adapted classical AL baselines.

**Strengths:**

- Thorough empirical validation—extensive ablations, layer-wise probes, traditional-AL comparisons, and robustness checks across diverse model families.

- Practical impact—simple MLP probe and T5-based decoder add negligible inference cost yet yield large savings, with reproducible code provided.

- Clear writing and well-motivated research questions.

**Weaknesses:**

- The novelty should be highlighted. Adding a small model for LLM active learning is not quite new. What are the most significant differences between the current method and the existing ones, e.g., FreeAL? The semantic entropy here is more like a prediction confidence, please refer to DeepConf (https://arxiv.org/abs/2508.15260) for the related work.

- Scope is limited to factual closed-book QA; unclear how well the probe transfers to open-ended or reasoning-heavy tasks.

- Generated QA pairs still need external annotation; the paper does not quantify human effort for this step.

**Questions:**

Please see the weakness.

---

### Official Review · Reviewer_5ogy · 2025-11-01

**Soundness:** 3
**Presentation:** 3
**Contribution:** 3
**Rating:** 4
**Confidence:** 3

**Summary:**

Summarize: This paper proposed the Knowledge-Aware Active Learning (KA2L) framework. It leverages the language model hidden states and the Semantic Entropy metric on sampled outputs to train a classifier that determines whether the model has the knowledge for a given question. The result shows that training on data with unknown questions is more effective, saving annotation and compute costs. The paper also employs a latent space decoding technique from LLM interpretability research to augment the unknown questions. KA2L also outperforms adapted classic active learning methods.

**Strengths:**

1. The idea of using semantic entropy to guide data selection in active learning is novel.

2. The result is validated with detailed experiments on nine open-source LLMs.

**Weaknesses:**

1. A core claim of this paper is that “a higher SE value suggests greater semantic divergence, indicating that the model has not mastered the knowledge associated with the question” (Sec 3.2.2), but there are no experiments explicitly validating that a higher SE score is directly related to a lower performance metrics, making this claim questionable.

2. In the experiments, this paper constructs a D_combine dataset “simulating a standard, unfiltered dataset collected without an active learning strategy” by mixing D_unk and D_k equally. The question is why not directly sample data from the original pool? D_unk samples and D_k samples may not be equally distributed.

3. The paper claims that KA2L can “cut annotation and computational costs by approximately 50% while maintaining high performance” (Sec 5.1) by comparing 5k Unknown and 10k Combine sets. This ignores the computation overhead of KA2L’s workflow: probing (training and inference) and possible augmentation. Claiming a napproximately 50% computational costs reduction could be an unfair comparison.

**Questions:**

The code repo link provided by the authors is not working as every file shows “The requested file is not found.”

---

### Note · Authors · 2025-12-01

I have read and agree with the venue's withdrawal policy on behalf of myself and my co-authors.